

# Synchronizing ice-core and U/Th time scales in the Last Glacial Maximum using Hulu Cave ¹⁴C and new ¹⁰Be measurements from Greenland and Antarctica

*Giulia Sinnl[1], Florian Adolphi[3], Marcus Christl[6], Kees C. Welten [4], Thomas Woodruff [5],*

*Marc Caffee [5], Anders Svensson[1], Raimund Muscheler[2] and Sune Olander Rasmussen[1]*

*[1] Physics of Ice, Climate, and Earth, Niels Bohr Institute, University of Copenhagen, Copenhagen, Denmark.*

*[2] Quaternary Sciences, Department of Geology, Lund University, Lund, Sweden*

*[3] Alfred Wegener Institute, Helmholtz Centre for Polar and Marine Research,*

*Bremerhaven, Germany*

*[4] Space Sciences Laboratory, University of California, Berkeley, California, USA*

*[5] Department of Physics and Astronomy, Purdue University, West Lafayette, Indiana, USA*

*[6] Laboratory of Ion Beam Physics, ETH Zurich, Zurich, Switzerland*

**Correspondence to:** giulia.sinnl@nbi.ku.dk

## Abstract

Between 15 and 27 ka b2k (thousands of years before 2000 CE) during the last glacial, Greenland experienced a prolonged cold stadial phase, interrupted by two short-lived warm interstadials. Greenland ice-core calcium data show two periods, preceding the interstadials, of anomalously high

atmospheric dust loading, the origin of which is not well understood. At approximately the same time as the Greenland dust peaks, the Chinese Hulu Cave speleothems exhibit a climatic signal suggested to be a response to Heinrich Event 2, a period of enhanced ice-rafted debris deposition in the North Atlantic. In the climatic signal of Antarctic ice cores, moreover, a relative warming occurs between 23 and 24.5 ka b2k that is generally interpreted as a counterpart to a cool climate phase in

the Northern Hemisphere. Proposed centennial-scale offsets between the polar ice-core time scales and the speleothem time scale hamper the precise reconstruction of the global sequence of these climatic events. Here, we examine two new ¹⁰Be datasets from Greenland (NorthGRIP) and Antarctic





(WDC) ice cores to test the agreement between different time scales, by taking advantage of the globally synchronous cosmogenic radionuclide production rates.

Evidence of an event similar to the Maunder Solar Minimum is found in the new [10]Be datasets, supported by lower-resolution radionuclide data from Greenland and [14]C in the Hulu Cave speleothem, representing a good synchronization candidate at around 22 ka b2k. By matching the respective [10]Be data, we determine the offset between the Greenland ice-core time scale, GICC05, and the WDC Antarctic time scale, WD2014, to be 125±40 years. Furthermore, via radionuclide

wiggle-matching, we determine the offset between the Hulu speleothem and ice core timescales to be 375 years for GICC05 (75–625 years at 68% confidence), and 225 years for WD2014 (-25–425 years at 68% confidence). The rather wide uncertainties are intrinsic to the wiggle-matching algorithm and the limitations set by data resolution. The undercounting of annual layers in GICC05 inferred from the offset is hypothesized to have been caused by a combination of underdetected annual layers,

especially during periods with low winter precipitation, and misinterpreted unusual patterns in the annual signal, during the extremely cold period often referred to as Heinrich Stadial 1.

## 1   Introduction

Paleoclimatic studies rely on climate records with accurate timescales to allow identification of the driving mechanisms of climate change. At present, numerous independent chronologies, based on

vastly different methods, are used to date specific climatic archives via, for example, layer counting, measuring the U/Th decay, or radiocarbon dating. The uncertainties and inaccuracies of each of those time scales are often difficult to assess and are major obstacles to an accurate and complete global reconstruction of paleoclimate.

The Last Glacial Maximum (LGM) was the last period when the ice sheets were at their largest extent

before the deglaciation into the Holocene, although its exact stratigraphic definition is being debated (Hughes et al., 2013). For terrestrial records of the Northern Hemisphere, Hughes & Gibbard (2015) suggested defining the LGM as the cold period Greenland Stadial 3 (GS-3: 23.3-27.5 ka b2k following Rasmussen et al., 2014), based on the analysis of NorthGRIP dust and sea-level records. In other ice-extent reconstructions, the LGM was established to have lasted until mid GS-2, around 20

ka b2k (Clark et al., 2009). In the present study, we refer to the period 20-25 ka b2k as the LGM for simplicity, since it coincides with the age limits of our new Greenland [10]Be dataset, being aware that this is not a formal stratigraphic definition.



During this time, a phase of massive discharge of icebergs from the Laurentide ice sheet was inferred from the ice-rafted debris content of North Atlantic marine sediments, defining the occurrence of the
Heinrich Event 2 (HE-2; Bard et al., 2000; Peck et al., 2006). HEs occurred during some GSs and added a large amount of freshwater to the surface ocean, likely causing a more extreme shutdown of the Atlantic overturning meridional circulation (AMOC) and, thereby, prolonged climatic conditions of extreme cold (McManus et al., 2004). The term Heinrich Stadial (HS) is often used to indicate the period affected by the HE. The duration of HS-1, for example, is limited to the 14.5-17.5 ka b2k
interval within GS-2.1 (Broecker and Barker, 2007), while for HS-2, a correspondence with the late GS-3 is often argued for, based on speleothem water isotope records (e.g. Li et al., 2021).

Associated with HS-2, one could expect a cooling signal in the stable water isotopes of Greenland ice cores ($\delta^{18}O_{ice}$; Jouzel et al., 1997) but this is not the case. Possible reasons include that the high-latitude areas around the ice sheet were not affected by the cooling or that the isotopes reacted non-
linearly (Guillevic et al., 2014). For example, it has been suggested that, during HS-1, the empirical correlation of temperature and $\delta^{18}O_{ice}$ was disrupted by precipitation-pattern alterations (He et al., 2021), which may also be the case for HS-2.

Mineral dust aerosols in Greenland ice cores (as reflected by e.g. $Ca^{2+}$ measurements) have their major source on the Eurasian continent and mainly reflect the dryness of the source region and wind
strength, which are highest in periods of extreme cold (Schüpbach et al., 2018). In addition to the already high stadial calcium levels, during GS-3 calcium deposition in Greenland shows two periods of further increased concentration (Rasmussen et al., 2014), which could be related to atmospheric reorganization following HE-2, although this attribution is still speculative (Hughes & Gibbard, 2015). Also, methane ($CH_4$) emissions originating from tropical wetlands and trapped in the ice cores
may have strengthened during HSs and could be used as another indicator of the HS-2 climate (Rhodes et al., 2015), since higher methane levels are found in Antarctic ice cores around the same time.

The objective of this study is a comparison of three time scales in the LGM. The time scales we are going to examine are the speleothem time scale from Hulu Cave, China (Wang et al. 2001; Southon
et al., 2012; Cheng et al., 2016), the time scale for the Antarctic WDC ice core (WD2014; Sigl et al., 2016), and the Greenland ice core chronology (GICC05; Andersen et al., 2006; Svensson et al., 2008). The Hulu Cave speleothems were dated by U/Th measurements for the period from 15 to 55 ka b2k and have previously been analysed for carbon isotopes ($^{14}C$; Southon et al., 2012; Cheng et al., 2018) and climatic proxies, such as stable oxygen isotopes ($\delta^{18}O_{calcite}$; Wang et al. 2001; Cheng et al., 2016).



The dead carbon fraction (DCF) of $^{14}C$ in speleothems from this cave, that is, the estimated amount of older carbon that may contaminate the recorded atmospheric $^{14}C$-signal, is very low and assumed to be constant over time (Cheng et al., 2018) making these speleothems excellent candidates for calibration of the IntCal20 curve (Reimer et al., 2020).

Throughout the 15 to 42 ka b2k period, the GICC05 was constructed by manually counting annual

layers in the electrical conductivity (ECM), continuous flow analysis (CFA) ion records, and visual stratigraphy of the NorthGRIP ice core (Andersen et al., 2006; Svensson et al., 2006). The uncertainty of GICC05 was assessed by identifying so-called 'uncertain annual layers', each contributing to the chronology and to the Maximum Counting Error (MCE) as 0.5±0.5 years (Rasmussen et al., 2006). The GRIP and GISP2 ice cores were tied to NorthGRIP by identification of volcanic tie points and

biomass burning events, and GICC05 was extended to these ice cores by interpolation (Rasmussen et al., 2008; Seierstad et al., 2014).

For Antarctica, the WD2014 time scale was constructed by manual and automatic counting of layers in the WDC ice core until 31.2 ka b2k (Sigl et al., 2016). The data used for counting were the ECM and CFA impurity records; however, for the period 15-27 ka b2k, only ECM data were suitable for

layer counting because of insufficient resolution of the CFA record. The uncertainty was assessed by comparison to the previous time scale (WAIS Divide Project Members, 2013), as well as by comparing the manual and automated versions of the layer count (Winstrup et al., 2012).

Around 23 ka b2k, the $\delta^{18}O_{ice}$ of Greenland indicates two short-lived Greenland interstadials, also known as Dansgaard-Oeschger events, namely GI-2.1 and 2.2 (Rasmussen et al., 2014). Antarctica

experienced warming between 24.5 and 24 ka b2k, as shown by the Antarctic Isotope Maximum 2 (AIM-2); subsequently, the water isotope levels remained high until ~23 ka b2k, after which a cooling trend lasted until around 22.2 ka b2k (EPICA Community Members, 2006). Greenlandic and Antarctic $\delta^{18}O_{ice}$ records are hypothesized to be coupled by the bipolar seesaw mechanism (Stocker & Johnsen, 2003; Pedro et al., 2018), connecting GI-GS pairs and AIM stages.

By synchronizing Antarctic $CH_4$ and Greenlandic $\delta^{18}O_{ice}$, the WAIS project members (2015) stated that, on average, the onset of Antarctic cooling lags the onset of the GI warming by 218±92 years. The authors duly excluded the GI-2−AIM-2 pair from their lead-lag analysis, firstly because the GISP2 $CH_4$ record did not support synchronicity with the GI-2 temperature increase, and, secondly, because the older HE-4 and HE-5 were similarly associated with higher $CH_4$ levels. Recently,

Svensson et al. (2020) presented a bipolar volcanic match between Greenland and Antarctic ice cores,





allowing re-synchronization of the GI-AIM pairs. They find an average delay of 122±24 years but the pair GI-2–AIM-2 is also excluded from their analysis.

According to the current chronologies, the onset of Antarctic cooling, i.e. the peak of the AIM-2 event, can be visually identified in the $\delta^{18}O_{ice}$ of WDC around 23.6 ka b2k (Jones et al., 2017), leading

the corresponding onset of GI-2.2 by about 260 years (Rasmussen et al., 2014; Sigl et al., 2016). Therefore, the pattern of the bipolar seesaw appears different for AIM-2, both because the lead-lag dynamic with Greenland appears to be reversed and because the AIM-2 phase has a disproportionate duration compared to the very short GI-2.1 and GI-2.2 interstadials. Resolving some time-scale issues, which we will delineate shortly, will clarify the distinctive timing factors of the global climate

around HE-2, compared to the 'conventional' bipolar seesaw scenario.

Traces of volcanic eruptions and cosmogenic radionuclides provide synchronization tools that do not rely on the precise identification of climatic match-points and on the assumption of their synchronicity. While aligning GIs across climatic archives provides a broad overview of climate in different regions, the assumed synchronicity of GIs prevents us from assessing the actual leads and

lags within the climate system. Moreover, the climatic tie points may be difficult to identify or not available in all time periods. For example, while $\delta^{18}O_{calcite}$ of Asian speleothems show a clear signal at the time of HE-2, consistent with large-scale climate changes (Li et al., 2021), there is no counterpart of the brief GI-2.1 and 2.2, hampering synchronization to Greenlandic $\delta^{18}O_{ice}$.

Across the LGM, most studies lack bipolar and inter-regional tie points to allow for an accurate

reconstruction of the sequence of events. For example, Svensson et al. (2020) do not report bipolar volcanic tie points over the entire period 16.5 to 24.7 ka b2k. At 24.7 ka b2k, they evaluate GICC05 to be ~85 years older than WD2014. Another inter-regional synchronization effort by Corrick et al. (2020) offers a climatic synchronization of speleothem and Greenland $\delta^{18}O$, but also lacks tie points over the LGM. Their estimation around GI-3 (27.78 ka b2k) is that GICC05 is 90 years younger than

U/Th-dated samples. A comparison between WDC $CH_4$, Hulu $\delta^{18}O$, and Greenlandic $\delta^{18}O$ (Sigl et al., 2016) found that, at the onset of GI-3, WD2014 is 167 years younger than the Hulu time scale and that GICC05 is younger than WD2014 by about 30 years and therefore 197 younger than Hulu. These two latter studies agree that around GI-3, events are younger according to the ice-core time scales than according to the U/Th time scales, albeit disagreeing about how much, while there is

indication that the offset between GICC05 and WD2014 changes sign between 27.78 and 24.7 ka b2k.



Given the scarcity of climatic and volcanic tie points over the LGM, in this work, we focus on new measurements of cosmogenic radionuclides to directly compare polar ice cores and the Hulu time scale. The interaction of galactic cosmic rays (GCRs) with the atmospheric parent atoms (N, O) of
$^{10}Be$ and $^{14}C$ is modulated by the time-varying helio- and geomagnetic fields. Therefore, the radionuclides recorded in climatic archives ideally show synchronizable features in their production history (Steinhilber et al., 2012; Adolphi et al., 2018). The $^{14}C$ atoms enter the carbon cycle, which causes delay and smoothing of the atmospheric $^{14}C$ concentration relative to the production signal. However, the connection with the rapidly deposited $^{10}Be$ in ice cores (1-2 years depositional delay;
Raisbeck et al., 1981) can be made using carbon-cycle models. The model we apply in this study was used extensively in works by Muscheler et al. (2000, 2004, 2009, 2014) and Adolphi et al. (2014, 2016, 2018) and is based on the box-diffusion model by Siegenthaler (1983). The main assumption that $^{10}Be$ varies proportionally to the true global production rate of cosmogenic radionuclides may, however, lead to uncertainties. For example, changes in the balance between wet and dry deposition
or changes in the transport of $^{10}Be$ to the ice sheet are possible factors that might alter the signal recorded in ice cores from the true production rate. Similarly, $^{14}C$ may be affected by changes in the carbon cycle, adding additional signals that are not related to production rate changes.

Currently, the only radionuclide-based tie point in the LGM between ice-core and Hulu Cave records was found by Adolphi et al. (2018). An offset of 550 years (95% probability interval: 215-670 years)
between GICC05 (younger) and the Hulu time scale (older) was estimated at 22 ka b2k, shortly after the GS-2.1 onset. Here, we test and refine this result by using new higher-resolution $^{10}Be$ data from Greenland and adding new Antarctic $^{10}Be$ data to our comparison. We also investigate if and where the ice-core time scales may have accumulated high dating inaccuracies and reconstruct the timing of events across the LGM.

**2    Data and Methods**

In this study, we aim at comparing the new cosmogenic radionuclide data with other datasets from Greenland, Hulu Cave, and Antarctica. We also analysed water stable isotope, methane, and calcium data to assess climatic changes. We summarize the relevant datasets, age resolutions, and citations in table 1.



### 2.1 Preparation and measurement of the NorthGRIP samples


The 322 new Greenland $^{10}$Be measurements were performed at ETH Zurich on samples from the NorthGRIP ice core between 1726.45 m and 1816.51 m depth, which according to GICC05 correspond to ages between 20039 to 24774 years b2k (Andersen et al., 2006). The samples have a variable temporal resolution between 7.5 years and 14 years with some smaller gaps (see Methods

Appendix, sec. 8.2) and are the so-far best-resolved available radionuclide dataset for the LGM. The measured $^{10}$Be concentrations are shown in fig. 1a.

*Table 1 Datasets used in this study. Age resolutions are calculated for the period 20 to 25 ka b2k. For the Hulu Cave, $^{14}$C data from two speleothems from the cave, labelled H82 and MSD, were published in two separate studies.*

| Dataset | Location | Proxy | Avg. Age Res. [years] | Reference |
|---|---|---|---|---|
| NorthGRIP | 75.10N 42.32W | $^{10}$Be | 10 | This study |
| | | $Ca^{2+}$ | 20 | Erhardt et al., 2021 |
| | | $\delta^{18}O$ | 10 | NorthGRIP members, 2004 |
| GRIP | 72.58N 37.64W | $^{10}$Be | 27 | Yiou et al., 1997; Muscheler et al., 2004 |
| | | $\delta^{18}O$ | 10 | Johnsen et al., 1997 |
| GISP2 | 72.36N 38.30W | $^{10}$Be | 162 | Finkel & Nishiizumi, 1997 |
| | | $\delta^{18}O$ | 10 | Stuiver & Grootes, 2000 |
| WDC | 79.46S 112.085W | $^{10}$Be | 67 | This study |
| | | $\delta^{18}O$ | 10 | Jones et al., 2017 |
| | | Non-sea-salt-Sulfate (nssS) | 1 | Buizert et al., 2018 |
| Hulu Cave | 32.5N 119.16E | $^{14}$C | 271 | H82: Southon et al., 2012 |
| | | | 124 | MSD: Cheng et al., 2018 |
| | | $\delta^{18}O$ | 70 | H82: Wang et al., 2001 |
| | | | 31 | MSD: Cheng et al., 2016 |





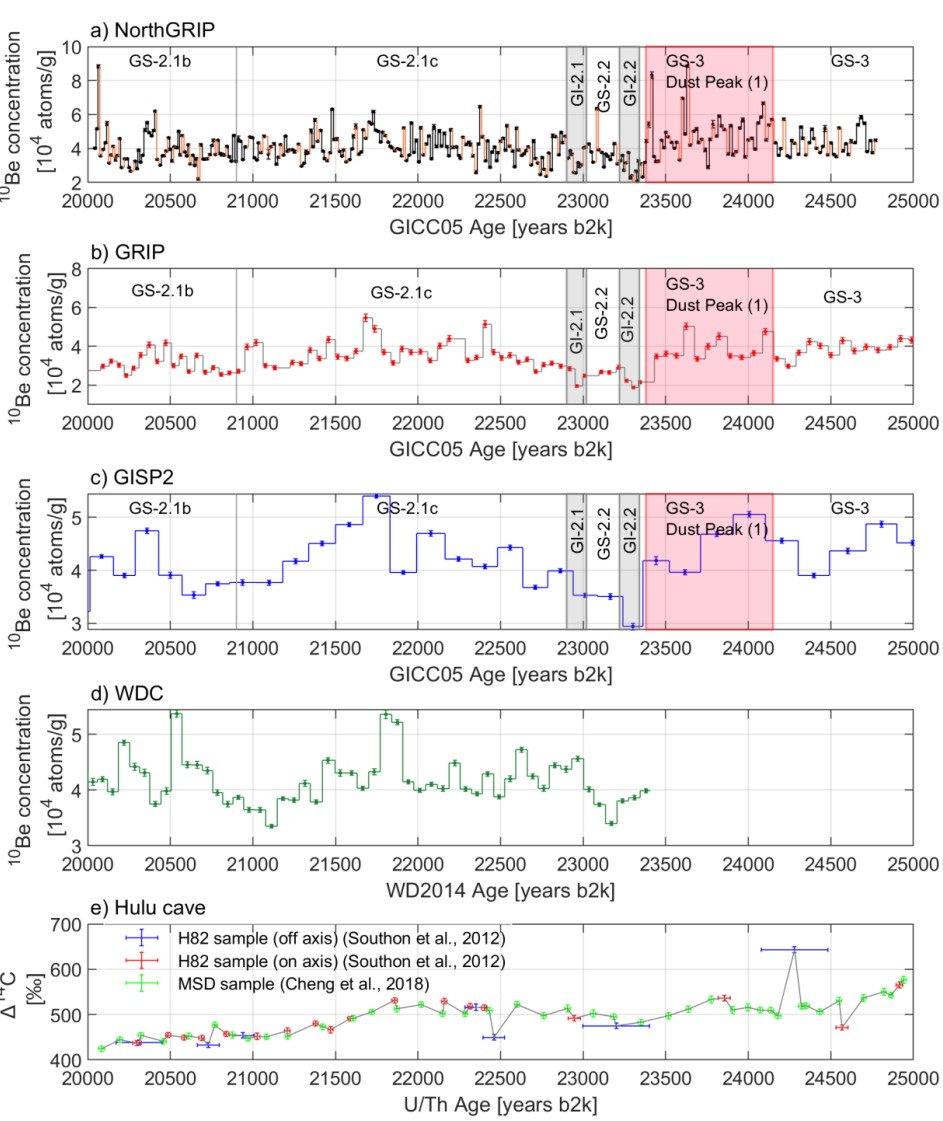


*Figure 1 Cosmogenic radionuclide data used in this study.*

*On the GICC05 timescale: (a) NorthGRIP $^{10}$Be concentrations (orange lines indicate discontinuities in data collection); (b) GRIP $^{10}$Be concentrations (Yiou et al., 1997; Muscheler et al., 2004). Data are available from every $2^{nd}$ ice-core portion of 55 cm, leading to discontinuities in the dataset between each 27-year resolved*

*data point (grey lines). (c) GISP2 $^{10}$Be concentrations (Finkel & Nishiizumi, 1997). On the WD2014 timescale: (d) WDC $^{10}$Be concentrations.*

*On the U/Th timescale: (d) Hulu Cave $\Delta^{14}C$ data as reported in three separate datasets. In the following, we refer to the H82 and MSD samples, but we exclude the off-axis H82 measurements (blue) by Southon et al. (2012), as they show more outliers and wider dating uncertainties.*



## 2.2 Preparation of the WDC samples and measurement

A total of 73 samples in the WAIS Divide 06A ice core (WDC-06A), from 2453 to 2599 m depth, were analysed for $^{10}$Be concentrations at Purdue University. These samples represent continuous ice-core sections with a cross-section of ~2 cm$^2$ and a length of ~2 m (varying from 1.89 to 2.12 m), corresponding to ~60-75 years of snow accumulation per sample. See Methods Appendix sec. 8.1 for more details. The measured WDC $^{10}$Be concentrations are shown in Fig. 1(d).

## 2.3 Conversion of $^{10}$Be concentrations to fluxes

To account for the first-order correction of climatic influences on the $^{10}$Be signal (Adolphi et al., 2018), $^{10}$Be concentrations need to be converted to fluxes, which requires knowledge of accumulation rates (see Methods Appendix sec. 8.3). Accumulation rates for the ice cores are reconstructed using the annual layer thicknesses and an appropriate model for the layer thinning. The thinning in the LGM portion of the ice cores can be approximated by a linear function of depth, although there may be uncertainties that relate to the time scale itself.

For NorthGRIP, the layer thickness is known by direct layer counting (Andersen et al., 2006), while the thinning was modelled by Johnsen et al. (2001). For GRIP and GISP2, the annual layer thickness is interpolated from the volcanic match to NorthGRIP (Rasmussen et al., 2008; Seierstad et al., 2014). During interstadials with well-resolved volcanic tie points, measurements indicate that the two Summit cores (GRIP and GISP2) have a stadial-to-interstadial accumulation increase that is 10% higher than at NorthGRIP (Seierstad et. al, 2014). Due to the scarcity of volcanic tie points across the LGM, the accumulation difference for GI-2.1 and 2.2 may be underestimated for GRIP and GISP2.

The most recent version of thinning functions for the GRIP and GISP2 was used to calculate fluxes in this study (Lin et al., 2021; Hvidberg et al., 1997). For WDC, the accumulation rate was reported by Fudge et al. (2016) inferred from the WD2014 layer thicknesses and modelled thinning (Buizert et al., 2015).

The fluxes of the new datasets are shown in Fig. 2, together with the modelled accumulation rates.

The fluxes of NorthGRIP were checked for residual correlation with climate proxies (fig. S4), showing that the flux conversion largely removes the climatic influence on $^{10}$Be deposition in Greenland.





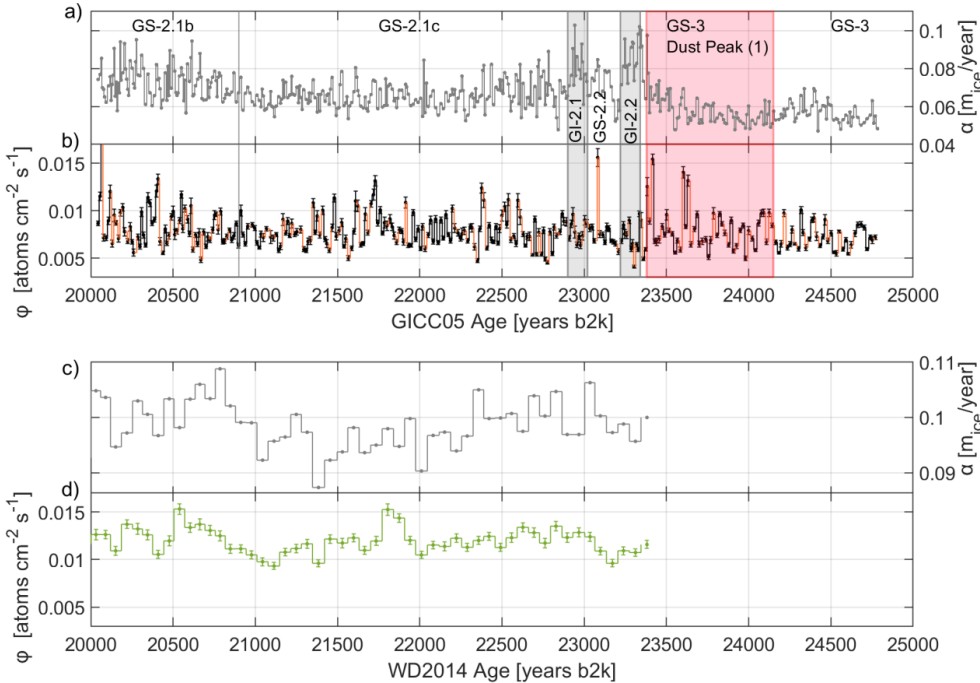

*Figure 2 Accumulation rates and ¹⁰Be fluxes of NorthGRIP (a, b) and WDC (c, d).*

*(a) The accumulation rate of NorthGRIP was obtained from the GICC05 layer thicknesses and the strain model (Johnsen et al., 2001). The accumulation rate was converted to the same depth resolution of each ¹⁰Be data point by averaging the accumulation rate between the top and bottom depths of each ¹⁰Be sample and determining the uncertainty from the standard deviation of the annual accumulation within each depth interval. (b) The ¹⁰Be fluxes of NorthGRIP show less climate-related fluctuations than the concentrations, as expected, and have an average of $0.008 \pm 0.002$ atoms cm⁻² s⁻¹. (c) The published accumulation rate of WDC (Fudge et al., 2016) was down-sampled to the same age resolution of the WDC ¹⁰Be concentrations. (d) The WDC ¹⁰Be fluxes are shown on the same scaling as the NorthGRIP fluxes for comparison. The average WDC flux is $0.011 \pm 0.001$ atoms cm⁻² s⁻¹, higher than in Greenland either because of depositional differences between the poles (Heikkilä et al., 2013) or because of accumulation rate inaccuracies.*

### 2.4 Carbon cycle modelling and uncertainties

A carbon-cycle model (here the box-diffusion model by Siegenthaler, 1983) is necessary to derive the atmospheric $\Delta^{14}C$ signal, i.e. the decay and fractionation-corrected ratio of $^{14}C/^{12}C$ relative to a standard (Stuiver & Pollach, 1977), from the measured ice-core ¹⁰Be. The model should be run with parameters that best represent the state of the carbon cycle and its changes through time, with the expectation that any remaining variability will be related purely to production effects.



To compare the measured and the modelled $\Delta^{14}$C, in this study we will make use of linear detrending, as this largely removes the systematic offsets associated with the unknown carbon cycle history and inventories. By varying the model parameters, some residual effect of the parametrization is observable in the amplitude of modelled $\Delta^{14}$C changes, but not in the timing of the changes, which is
most important here.

In the period 20-25 ka b2k, the Hulu Cave $\Delta^{14}$C is about 500 ‰ (fig. 1e), which is higher than early Holocene values, which are below 200 ‰ (Reimer et al., 2020). These higher values may be related to one or more of the following factors: a lower ocean diffusivity during the LGM or any process that similarly reduces the carbon uptake by the ocean (Muscheler et al., 2004), a lower atmospheric $^{12}$C
inventory resulting in higher $^{14}$C/$^{12}$C ratios (Köhler et al., 2022), or a weaker geomagnetic field. For instance, the $^{10}$Be production rates in the LGM are expected to have been about 20% higher than today due to the lower geomagnetic field intensity during the LGM (Muscheler et al., 2004).

The strength of the geomagnetic field directly affects both the $^{10}$Be and $^{14}$C production rates. Although each radionuclide may be affected differently (Masarik & Beer, 2009), most studies do not find any
significant difference in production rates (e.g. Kovaltsov et al., 2012; Herbst et al. 2017). Adolphi et al. (2018) also showed that around a change in the geomagnetic field, $^{10}$Be production rates should be amplified by 30% to match the amplitude expected from geomagnetic field reconstructions. This could be explained by incomplete atmospheric mixing of $^{10}$Be as the geomagnetic shielding effect of GCR's is largest at the equator.

To determine the most appropriate model parameters, we repeat the calibration by Adolphi et al. (2018) around the Laschamps geomagnetic excursion at 41 ka b2k, since the available $^{14}$C data has been updated since then (Reimer et al. 2020). We run the model with different ocean ventilation values (fig. S2), finding that, for values of ocean diffusivity between 25-40% of the pre-industrial Holocene value, the modelled $\Delta^{14}$C match the IntCal20 data best. Moreover, this agrees with
Muscheler et al. (2004) who performed a time-dependent adjustment of the ocean diffusivity parameter between 10-25 ka to match the model to the measured $\Delta^{14}$C. In their study, the LGM ocean diffusivity was set to ~1000 m$^2$/yr, about 25% of the pre-industrial Holocene value.

We find that a 20% production rate amplification of the normalized $^{10}$Be and an ocean diffusivity of 25% of the pre-industrial Holocene value produce modelled outputs of about 500 ‰, in agreement
with the Hulu-cave measurements, although the model fails to capture the decreasing trend. We note that this is not necessarily a realistic parameter of the state of the carbon cycle, but allows us to match





some of the main features seen in the data. We associate no uncertainty with the model parameters since no setup realistically explains all the $\Delta^{14}$C features.

### 2.4.1 Sensitivity tests: ocean diffusivity changes, accumulation rate uncertainties, measurement uncertainties.

We performed three separate sensitivity tests to provide an uncertainty boundary for the detrended modelled $\Delta^{14}$C curves. As a first test, we investigated how short-term changes in the ocean diffusivity affect the modelled output because during the LGM there likely have been changes in the carbon cycle around the GIs and the HS-2 (Bauska et al., 2021).

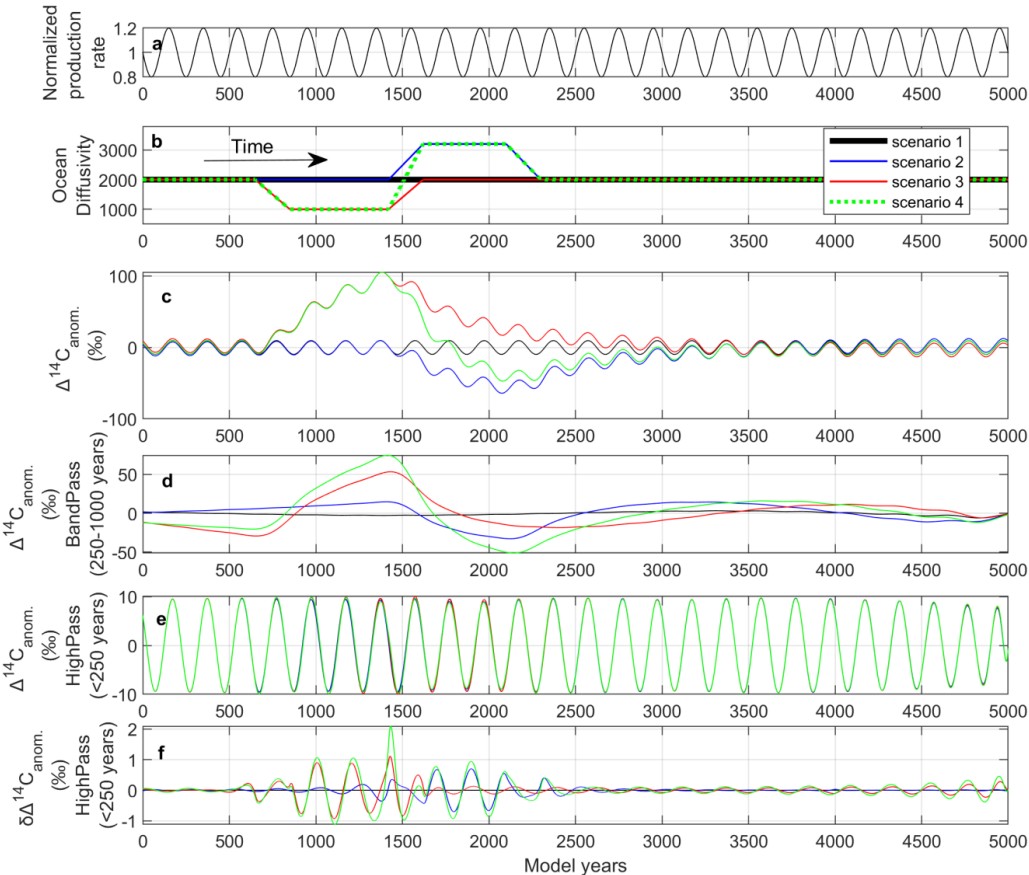

*Figure 3 Sensitivity tests for time-dependent ocean diffusivity changes.*



*(a) Normalized production rate input showing a cycle with an amplitude of ±25% and a period length of 200 years broadly consistent with typical solar de-Vries cycle variability as observed in $^{10}$Be data (e.g. Wagner et al., 2001). (b) Scenarios of ocean diffusivity, as described in the text. (c) Modelled output, after linear detrending. (d) Long-term variations of the output (band-pass filtered output with periodicity 250-1000 years).*
*(e) High-pass filtered output, up to 250-year periodicity, having amplitudes of around 20 ‰. (f) Differences of the high-pass filtered curves in (e) between the control scenario 1 and the other three scenarios (similar as in Adolphi & Muscheler, 2016). Differences of around 1-2‰ demonstrate that there is little influence of the ocean diffusivity scenarios on the high-pass filtered outputs of the model.*

A time-dependent change in ocean diffusivity was induced using 3 scenarios, as shown in fig. 3b:
either an abrupt increase in ocean ventilation (higher diffusivity: 80% of the pre-industrial value), an abrupt decrease in ocean ventilation (lower diffusivity: 25% of the pre-industrial value), or a sequence of both. The control scenario is set at 50% of the pre-industrial value. The duration of the events is chosen to reflect changes in the NorthGRIP calcium record, while the transition time was set to 200 years. The input signal is a 200-year periodic wave with similar values to the NorthGRIP normalized
$^{10}$Be concentrations.

Figure 3c shows that the effects of the perturbations in ocean diffusivity on $\Delta^{14}$C are quite high and span several tens of ‰, after detrending. Thus, any feature in the $^{14}$C records that is in the proximity of an abrupt climate change and has a comparable duration is uncertain and should not be used for matching. On the other hand, fig. 3d shows that short-term variations of the modelled $\Delta^{14}$C signal are
less affected by the diffusivity perturbation. At least in principle, signals exceeding 10-20 ‰ that are much shorter than the climatic transitions could be used for wiggle-matching, since fig. 3f shows that the different ocean diffusivities scenarios affect the high-passed $\Delta^{14}$C only within 2 ‰, hardly above measurement uncertainties.

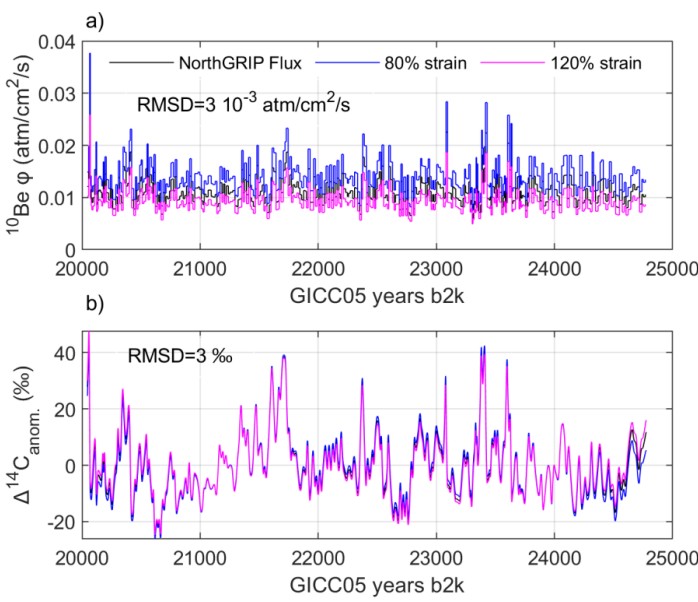

*Figure 4 Sensitivity test of the effects of strain-model related accumulation-rate uncertainties on the carbon-cycle modelling.*
*(a) Two strain-model scenarios produce different flux values. The RMSD quoted in the figure represents the average distance of the curves from each other. (b) The modelled $\Delta^{14}C$ curves are detrended, which largely removes the differences between the scenarios. However, the remaining variability is represented by the RMSD*
*between the curves, 3 ‰, which can be taken as the uncertainty derived from the ±20% perturbation of the ice accumulation model.*

A second sensitivity test was performed to investigate the effect of accumulation-rate uncertainties related to the strain model. We performed 2 experimental model runs where we shifted the thinning function (inverse strain) by ±20% of the mean value between 20 and 25 ka b2k, which is a realistic
modelling uncertainty between independent studies of the accumulation rate (Rasmussen et al., 2014; Gkinis et al., 2014). Changing the strain rate creates an uncertainty of about 3 ‰ between the modelled and detrended $\Delta^{14}C$ curves, as shown in fig. 4b. Furthermore, with a similar approach, we quantify the impact of $^{10}$Be measurements uncertainty on the modelled $\Delta^{14}C$ to be 1 ‰ (fig. S3). Adding these independent contributions in quadrature, we set an uncertainty for our modelled $\Delta^{14}C$
of 5 ‰. This uncertainty serves as an initial parameter for the wiggle-matching algorithm, described in the next section; it furthermore agrees with the uncertainty adopted for the comparison of centennial variations of $^{14}$C and $^{10}$Be during the stable climate of the Holocene (Adolphi & Muscheler, 2016). However, as we have verified with our test, $\Delta^{14}C$ changes in the vicinity of climate perturbations bear a considerably higher uncertainty.



## 2.5 The wiggle-matching algorithm reproduced from Adolphi & Muscheler (2016) and its uncertainty

Along with the visual inspection, an important tool for the quantification of offsets between timescales is the wiggle-matching algorithm, adapted by Adolphi & Muscheler (2016) from the original formulation by Bronk Ramsey et al. (2001), and described in detail therein. The first input for the algorithm is the detrended $\Delta^{14}$C as modelled from the ice-core $^{10}$Be concentrations or fluxes. The second input is the detrended $\Delta^{14}$C data from the Hulu Cave stalagmite samples. The output of the algorithm is a probability density function of the possible timescale offsets $\vec{t}_{offset}$.

Following the approach by Adolphi & Muscheler (2016), we investigated the probability within partially overlapping time windows $\overrightarrow{W}$, spaced every 50 years, obtaining a two-dimensional probability matrix $\boldsymbol{P}(\overrightarrow{W}, \vec{t}_{offset})$. This is intended as a way to study the offset in a time-dependent fashion, since the algorithm does not allow for stretching of the underlying timescales. We summarize the algorithm settings in the Method Appendix.

The position of the time-dependent mode of $\boldsymbol{P}$ is here defined to be the most likely timescale offset, $t^*_{offset}(\overrightarrow{W})$. By observing whether this latter quantity is stable and weakly time-dependent, we can visually mark some T1 and T2 limits for the purpose of producing an average offset within a subset of windows $\overrightarrow{W}$. An estimate of the average offset was thereby produced for the GICC05 and the WD2014 timescales.

After shifting each ice-core dataset by the proposed offset, we computed the $\chi^2$ test between the speleothem and ice-core $\Delta^{14}$C curves. If the p-value was outside the 0.01-0.99 interval, we repeated the wiggle-matching using the standard deviation of the residuals (RMSD in ‰) as the uncertainty for the modelled $\Delta^{14}$C curve. We then repeated the wiggle matching again and plotted the new ice-core $t^*_{offset}$ curves. We re-evaluated the boundaries T1 and T2, and re-averaged the $t^*_{offset}$ curves, obtaining a second estimation for the offset.

For this second offset, we propose an uncertainty estimation which considers the timescale uncertainties of both ice cores and speleothems, as well as resolution limitations. We consider the information gained by running the wiggle matching with multiple ice-core $^{10}$Be datasets, including concentrations and fluxes. Moreover, we consider the time-dependent behavior of the uncertainties. We therefore applied a Monte-Carlo protocol to estimate the uncertainty and conducted the following steps:



1) For each modelled $\Delta^{14}C$ dataset and window $W^*$, we approximate the $\boldsymbol{P}(W^*, \vec{t}_{offset})$ as a Gaussian distribution centered around the mode and with $1\sigma$ as the average lower and upper width at half maximum of $\boldsymbol{P}(W^*, \vec{t}_{offset})$. By arbitrarily choosing the mode of $\boldsymbol{P}$ as the best offset, we can disregard any other lobe of probability as spurious or unnecessary, with some confidence given by our visual inspection of the data.

2) For each dataset and window $W^*$, we sample randomly from the Gaussian for 10000 times to derive an ensemble of timescale offsets;

    3) By iterating steps 1 and 2 across all datasets and all windows within the established time boundaries T1 and T2, we compute the overall histogram of the sampled offsets;

    4) We evaluate the 68% confidence interval of this histogram, around the best offset established 370    as the mode.

This procedure is repeated separately for WD2014 and GICC05, and also for the Hulu H82 and MSD datasets. The algorithm is made available in the Supplement.

## 3   Results

### 3.1   A promising inter-ice-core tie point for $^{10}$Be synchronization

Previously used for the matching by Adolphi et al. (2018), a $^{10}$Be increase at 21.7 ka b2k (GICC05 age) is visible in the new NorthGRIP data as well as in the WDC dataset (fig. 1). This radionuclide increase resembles the $^{10}$Be signal associated with the Maunder Solar Minimum (1645-1725 CE), which was a period of low solar activity and consequent increase in the global radionuclide production (Berggren et al., 2009; Eddy, 1976). In figure 5, we compare the Holocene and LGM counterparts of 380   $^{10}$Be at NorthGRIP finding similar shapes and duration, which supports the attribution of the $^{10}$Be increase to a solar minimum during GS-2, which could explain the co-registration across $^{14}$C and $^{10}$Be datasets.

In the Holocene, high accumulation rates make the wet deposition of $^{10}$Be predominant over dry deposition, so concentrations may be more representative of the true $^{10}$Be production rate (Berggren 385   et al., 2009). During the glacial, model runs (e.g. Heikkilä & Smith, 2013) suggest that wet deposition is still predominant, but the abrupt accumulation rate changes observed across GIs require examining the fluxes to separate dilution effects from production effects. We examine both concentrations and





fluxes in fig. 5, finding that for both fluxes and concentrations, the LGM and Holocene signals are very similar.

Based on the observed similarities in fig. 5, we call the 22.7 ka b2k increase the "GS-2.1c $^{10}$Be Event" (abbreviated *G2B event* in the following) without claiming a certain solar origin of the signal. To support the bipolar synchronization, after resampling the NorthGRIP $^{10}$Be data on the resolution of WDC, we observe that the two ice cores register similar $^{10}$Be amplitudes around the G2B event. The flux increases by 0.003 atoms cm$^{-2}$ s$^{-1}$ at both sites, which represents an increase of 40% and 30%,

respectively, from the flux average values at NorthGRIP and WDC. The $^{10}$Be concentration, on the other hand, increased only 30% (1.3 x 10$^4$ atoms/g) and 20% (0.9 x 10$^4$ atoms/g), respectively, from the concentration average values at NorthGRIP and WDC.

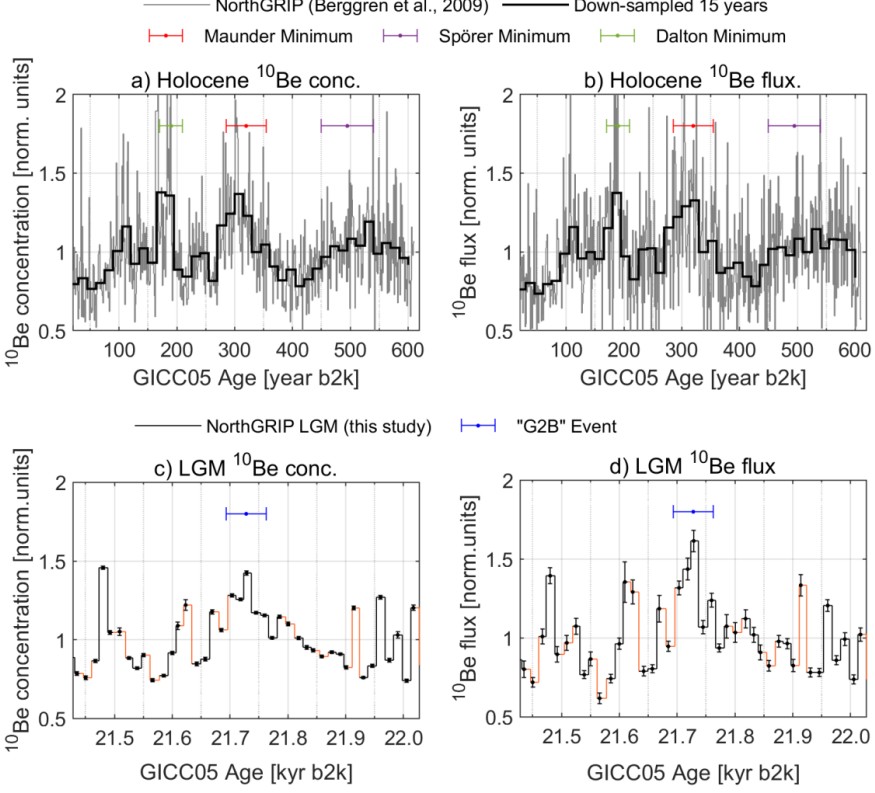

*Figure 5 Comparison of $^{10}$Be ice-core data in the Holocene, around the Maunder Minimum, and in the LGM,*
*around the G2B event.*





*By normalizing all data (dividing by their mean) within ±300 years of the central event, we ensure a better visual comparison of the data. (a, b) In the Holocene, the NorthGRIP data by Berggren et al. (2009) are compared to the defined durations of the grand solar minima, which were independently recorded in the sunspot observations. The down-sampling to 15 years allows for easier comparison to the LGM dataset below.*

*We can visualize the effect of solar minima as an increase in the $^{10}$Be production rate by about 40-50% from the average. The fluxes show a more abrupt increase in time, while concentrations record a more gradual increase. (c, d) In the LGM, the similarity of shape and duration to the Maunder Minimum supports the identification with a solar minimum.*

### 3.2 Synchronization between ice cores using $^{10}$Be

Having established the G2B event as a radionuclide production feature, we can synchronize Greenland ice cores by inserting new $^{10}$Be tie points between NorthGRIP and the GRIP and GISP2 ice cores. Furthermore, the G2B event can be used to improve the bipolar matching to Antarctica (fig. 6).

We decided to compare the $^{10}$Be data on similar resolutions to facilitate the identification of other

common production features. Hence, we down-sampled the high-resolution data of NorthGRIP to the same resolution as GRIP and WDC, as ice-core $^{10}$Be measurements are averages over the sampling depth intervals and lose variability with decreasing resolution.

*Table 2 $^{10}$Be tie-point ages between NorthGRIP, GRIP, and WDC. The internal difference between the Greenland ice cores (δ) reaches 27 years at the G2B event. Likewise, the difference between WDC and*

*NorthGRIP ages (Δ) indicates older ages for WDC at the G2B event and the youngest no. 4 tie point.*

| tie point | GICC05 Age (years b2k) | | | WD2014 Age (years b2k) | |
|---|---|---|---|---|---|
| | NorthGRIP | GRIP | δ | WDC | Δ(WDC-NorthGRIP) |
| 4 | 20373 | 20370 | 3 | 20541 | 168±40 |
| 5 | 21372 | 21347 | 25 | | |
| 6 | 21484 | 21458 | 26 | | |
| 7 (G2B event) | 21710 | 21683 | 27±21 | 21835 | 125±40 |
| 9 | 22383 | 22398 | -15 | | |
| 14 | 24119 | 24103 | 16 | | |

In figure 6, the $^{10}$Be concentrations from NorthGRIP, GRIP and WDC are shown on their respective time scales together with a set of published non-climatic tie points (Seierstad et al., 2014; Svensson et al., 2020). Between NorthGRIP and GRIP, we observe important similarities in the $^{10}$Be data,

which leads us to suggest 6 new $^{10}$Be tie points: a peak at 20.4 ka b2k, a double peak at 21.5 ka b2k,





the G2B event at 21.7 ka b2k, a single peak at 22.4 ka b2k, and a triple peak structure between 23.5 and 24.2 ka b2k. These tie points cover the previously tie-point-free section across GS-2.1b/c.

The match to WDC, of which the oldest tie point (no. 15) was published by Svensson et al. (2020), is extended with the aid of two additional $^{10}$Be tie points (no. 4 and no. 7/G2B). The choice of no. 4 as
a bipolar tie point is motivated by a similar layer count of ~1300 years from G2B and a similar shape of the signals.

The ages of the tie points are summarized in table 2 and derived from the mid-depth of the highest peak. The timescales of GRIP and NorthGRIP are slightly misaligned between 21 and 23 ka b2k by up to 27 years at the G2B event, probably due to the fact that GRIP ages were interpolated between
widely spaced tie points. The uncertainty of this misalignment can be estimated as half the sum of the resolutions of NorthGRIP and GRIP measurements ($\pm$ 21 years, 1 $\sigma$), as this is likely to limit our matching precision in each direction. Therefore, $^{10}$Be measurements cannot be said to resolve matching issues between Greenland ice cores with very high precision, nonetheless we will use the new $^{10}$Be tie points in the following to produce an updated accumulation rate for GRIP.
Furthermore, WD2014 and GICC05 are misaligned by 125$\pm$40 years at the G2B event, WD2014 being older; the uncertainty on this offset is given as half the sum of the resolutions of WDC and NorthGRIP data.





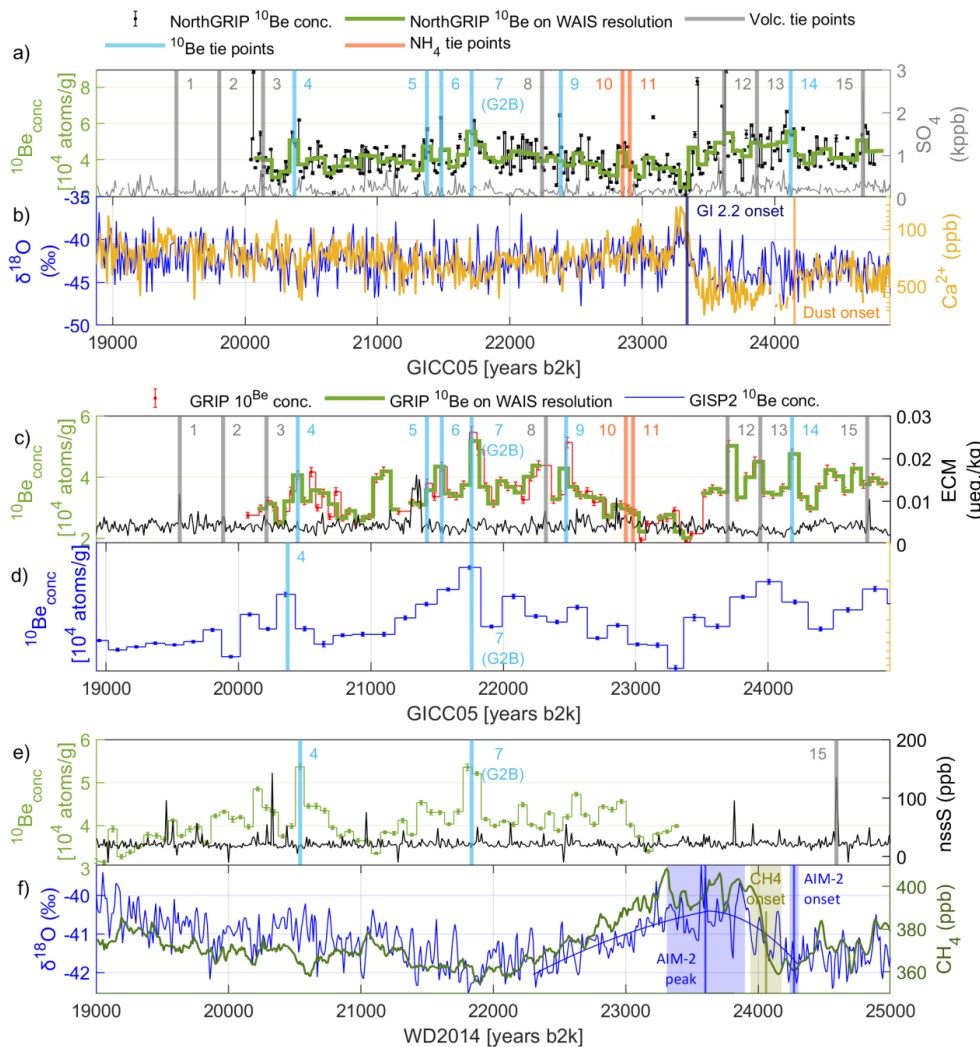

*Figure 6 Bipolar tie points and climatic proxies. Vertical bars indicate tie-point positioning.*

*The data were aligned using tie point 7 (G2B event), without stretching the timescales. (a) The NorthGRIP*
*$^{10}$Be concentrations (black) were down-sampled to the WDC resolution (green). The sulfate data (grey)*
*supports the volcanic match, while ammonium data, on which tie points no. 10 and 11 were based, are not*
*shown for clarity. (b) $\delta^{18}O$ and $Ca^{2+}$ at NorthGRIP (calcium is on an inverted log-scale) qualitatively represent*
*the climate and determine the timing of the GIs and the GS-3 dust peak.*



*(c) The GRIP $^{10}Be$ concentrations (red), having a 27-year resolution, were down-sampled to the WDC resolution (green). ECM (black) shows volcanic tie points, although some eruptions are better visible in the sulfate signal (not shown). (d) The $^{10}Be$ data of GISP2 (blue) has a low resolution, hence the alignment with NorthGRIP cannot be improved. Tie points no. 4 and 7 are however sufficiently visible in the data. (e) WDC $^{10}Be$ data (green) and nssS (black; no. 15 by Svensson et al., 2020). (f) WDC climatic proxies, with $CH_4$*

*presented on the gas chronology by Buizert et al. (2015). We observe the occurrence of the AIM-2 warming in Antarctica as an increase of $\delta^{18}O$ (blue). The approximate shape of the AIM-2 was calculated by a second-order polynomial fit. The AIM-2 onset and peak are characterized in the text. The age of the increase of $CH_4$ by ~50 ppb (orange) was calculated by detecting where the signal increased significantly above the mean, with its gas-age uncertainty (green shading, see text for details).*

To improve the GRIP time scale, we calculated a new depth-depth interpolation between the two Greenland ice cores and we obtained a timescale correction, which we apply in the following to the GRIP data. On the resolution of GISP2, $^{10}Be$ appears to be sufficiently aligned around the G2B event and the peak at 20.4 ka b2k, hence no time scale correction was applied to GISP2.

The fact that the G2B event is 27 years older in NorthGRIP than GRIP also means that the GRIP

accumulation rate needs to be corrected because of the corresponding changes to the layer thicknesses induced by the new tie points. This was done by multiplying the GICC05 timescale of GRIP by a correction function that computes the relative change of the layer thickness between the tie points. For example, since NorthGRIP has 1235 layers between tie-points nr. 4 and 7, while GRIP only has 1210 layers, the correction factor for the accumulation rate between these tie-points is 1.02.  Due to

these relatively small corrections, we do not find it necessary to apply any smoothing to the correction function. In the following, we denote fluxes of GRIP that were corrected in this way as "corrected fluxes" when we need to distinguish them from the previous version used by Adolphi et al. (2018). Although the effect of the accumulation correction is small, in section 3.3 we will show that this method has a significant impact on the amplitude of the G2B event signal in GRIP, making the

NorthGRIP and GRIP modelled $\Delta^{14}C$ curves look more alike.

In figure 6, panels b and f, selected climatic proxies are shown to illustrate that a bipolar match across the LGM is already sufficient to change the timing between the two GIs and the AIM-2, bringing the AIM-2 to be closer to the GI-2.2 onset (a "before version" of the alignment of the climatic proxies can be found in fig. 7).

In the WDC isotope data, we fit a curve to the $\delta^{18}O$ record over the AIM-2 period to determine both the likely age for the onset of the warming slope and the peak – all on the original WD2014 time scale. We used the Matlab function 'WDC_breakpoint' provided by WAIS project members (2015), which fits the AIM with a double-polynomial fit to identify the maximum. However, the fit is





sensitive to the starting guess for the position of the maximum; by varying the starting guess between
23 and 25 ka b2k, we observed that the fit finds several maxima. This fact is attributed to the shape
of the signal being ambiguous and very broad. Moreover, a visual maximum of the AIM-2 is clearly
identifiable at 23.6 ka b2k. Taking this as the central estimate, from the distribution of the fitted
breakpoints, we obtain the location of the AIM-2 peak as $23600 \pm 300$ years b2k ($1 \sigma$). On the other
hand, the onset of the AIM-2 is more precisely defined to be $24272 \pm 35$ years b2k ($1 \sigma$). We also
determine the onset of the $CH_4$ increase at $24060 \pm 118$ ($1\sigma$) years b2k, as the instant at which the
signal becomes higher (and remains higher) than 1 standard deviation from the average baseline of
the period 24.5-25.5 ka b2k. The uncertainty of the $CH_4$ onset is quoted from the gas age uncertainty
of WD2014 (Fudge et al., 2017), which is mostly useful to compare $CH_4$ to the $\delta^{18}O$ signal of the
same ice core, WDC. All values are summarized in table 3.
By aligning the two ice-core time scales at the G2B event, and without stretching the time scales, we
shift the GICC05 data by 125 years compared to WD2014 data. We observe that the AIM-2 peak now
happens 125 years before the onset of GI-2.2, instead of 250 years before as was the case on the
original time scales. The alignment at the G2B event is not sufficient to alter the order of the GI-2–
AIM-2 sequence. What the new alignment shows is that the AIM-2 warming occurs within the dust
peak and that the AIM-2 cooling is starting close to the dust peak termination in Greenland, with
potential influence of the GIs.

Finally, an average of the $^{10}Be$ fluxes of the three Greenland ice cores was calculated by stacking the
NorthGRIP, GRIP (corrected flux) and GISP2 fluxes, using Monte-Carlo bootstrapping (Adolphi et
al., 2018). For each iteration, three of the data series are selected with resampling, each dataset is
perturbed within its uncertainties, and averaged. The stack is shown in fig. S1, with uncertainty bands
derived from the standard deviation of the 1000 simulated fluxes.

The assumption behind averaging fluxes is that local accumulation effects are mostly removed by the
conversion to fluxes and that the climatic effects on $^{10}Be$ deposition are the same over Greenland.
Stacking the fluxes combines the information from three ice-core locations and therefore results in a
different $\Delta^{14}C$ than the interpolation over NorthGRIP data gaps.

### 3.3   Carbon cycle modelling and wiggle-matching

Inspection of the measured $\Delta^{14}C$ (fig. 7a) and the $^{10}Be$-based $\Delta^{14}C$ (fig. 7 b, c, e) confirms the
expectation that a shift of the two ice-core timescales towards older ages is required for a better
alignment with the Hulu-cave $^{14}C$ record.





We observe two non-climatic tie-points between all $\Delta^{14}$C curves:

1) The G2B event: a relatively abrupt increase of 30 ‰ in the modelled $\Delta^{14}$C from $^{10}$Be, reaching its maximum at 21,725 years b2k (GICC05 ages), about 100 years after the maximum is reached in $^{10}$Be fluxes. The most likely equivalent of this event is observed in the Hulu Cave data at around 22,200 years b2k (U/Th ages), where an increase of about 25 ‰ happens

abruptly between two data points which is followed by a slow decrease. In the H82 data by Southon et al., the increase is much less abrupt and spans 4 data points.

2) A smaller peak is observed in the ice-core data at 20,400 years b2k (GICC05). In Hulu Cave data, one elevated data point at 20,800 years b2k (U/Th) is visible in the MSD dataset, but the H82 record does not show this peak. In the original $^{10}$Be fluxes, this event is probably caused

by the combination of two peaks, one at about the same age and one at 20,600 years b2k (e.g., see fig. 2). The amplitude of the event is about 40 ‰ in the modelled $\Delta^{14}$C.

We assume that the modelled and the measured $\Delta^{14}$C should, after detrending, be dominated by the same production signal and that the differences we observe in the datasets can be explained by noise, modelling uncertainties, and dynamics affecting the deposition rather than the production of the

radionuclides. Therefore, we proceed with the wiggle-matching despite the observed differences. Based on these considerations, we run the wiggle-matching algorithm exclusively in the 20.5-22.5 ka b2k range (U/Th ages), using data windows of 1300 years with an overlap of 50 years between successive windows. The 1300-year choice is motivated by a compromise between enough signal inclusion in each window and more precision, since preliminary tests showed that this value returned

the most stable results, compared to windows of 1000 or 1500 years. For the U/Th time scale, we use both Hulu datasets but we separate our treatment for the H82 and the newer MSD data (Southon et al., 2012; Cheng et al., 2018), because of the different resolution and the slightly different signals recorded. For the WD2014 timescale, we use both the WDC $^{10}$Be concentrations and fluxes, modelled to $\Delta^{14}$C, since they are similar after carbon modelling. For the GICC05 timescale, we use all available

$\Delta^{14}$C modelled from concentrations, fluxes, and the flux stack, since they are similar at least around the production features we are going to match.

By averaging the obtained offset curves across the datasets and the interval (fig. S5), we estimate the initial time scale offset to be 370 years for GICC05 and 225 years for WD2014, using the MSD dataset exclusively. We do not give an uncertainty boundary for these offsets just yet, as we first

perform a $\chi^2$ test, following the protocol outlined in the Methods section. We evaluated the $\chi^2$ and the



associated p-value between each shifted $\Delta^{14}$C curve and the Hulu dataset. For the flux-based datasets, the p-values were within the 0.01-0.99 threshold, except for the GRIP 'uncorrected' fluxes. This last exception motivates the exclusion of 'uncorrected' fluxes in the following, as we do not need to keep both GRIP flux-based curves, once we have ascertained that the 'corrected fluxes' satisfy our $\chi^2$ test,

possibly because of the improved chronological spacing of the samples.

In the case of all Greenlandic concentration-based curves, because of very low p-values of the $\chi^2$ (p<<0.01), we decided to set the $\Delta^{14}$C uncertainty to the RMSD of each dataset, which are between 12 and 21 ‰, less than the amplitude of the matched features. We repeat the wiggle-matching for





these cases and base our offset estimation on the full Greenland dataset with the uncertainty method

outlined above.

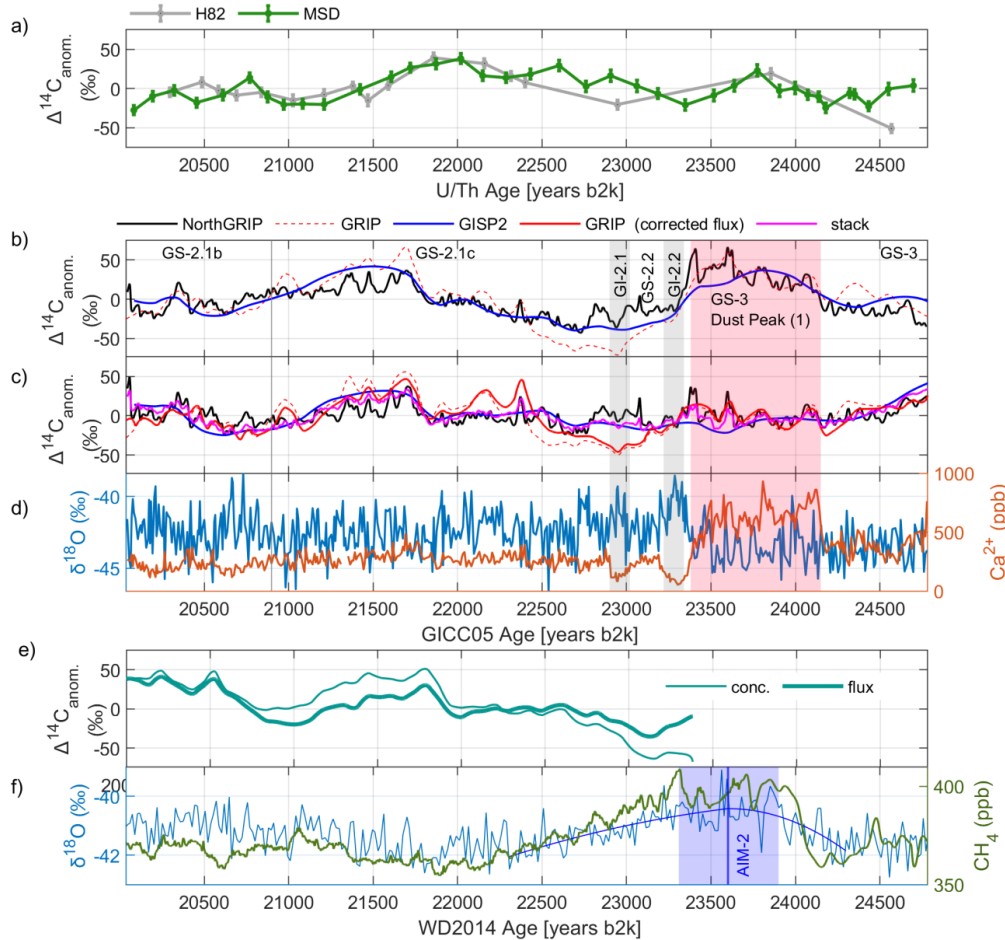

*Figure 7 Carbon-cycle modelled $\Delta^{14}C$ compared to measured Hulu Cave data and climatic data before synchronization.*

*(a) Measured Hulu Cave used for synchronization (H82: Southon et al., 2012; MSD: Cheng et al., 2018).*
*(b) Modelled $\Delta^{14}C$ from Greenland $^{10}Be$ concentrations (NorthGRIP, GRIP and GISP2 datasets).*
*(c) Modelled $\Delta^{14}C$ from Greenland $^{10}Be$ fluxes, including the flux stack (magenta). The effect of the flux correction of GRIP is mostly visible around the G2B event, where the corrected data shows a better agreement with the other cores. (d) The climatic data from NorthGRIP ($\delta^{18}O$ and calcium) contain the*
*signature of the dust peak and the GIs. (e) $\Delta^{14}C$ modelled from WDC $^{10}Be$ fluxes and concentrations. (f) Climatic data from the WDC core ($\delta^{18}O$ and CH_4).*



We therefore repeat the averaging of the timescale offset (fig. 8) between the MSD dataset and the GICC05 ensemble and we apply the Monte-Carlo iteration outlined in the Methods, between 21 and 22.1 ka b2k (the range highlighted in yellow in fig. 8a, where edge effects can be avoided). We obtain

the offset estimate of 375 years and a 68% confidence interval from 75 to 625 years. One assumption behind this approach is that the offset should only change slowly with time, which is supported by the flatness of most curves in fig. 8 (with some exception by the concentration-based offsets of NorthGRIP and GISP2).

For the WDC datasets, both flux- and concentration-based curves were shifted by 225 years. Upon

performing the $\chi^2$ test against the Hulu Cave data, the p-values were within the 0.01-0.99 tolerance interval, hence the uncertainty of the $\Delta^{14}C$ data did not need to be re-evaluated. We compute the average timescale offset (fig. 8) between the MSD dataset and the WD2014 ensemble of flux and concentration-based curves and we apply the Monte-Carlo iteration outlined in the Methods, between 20.8 and 21.75 ka b2k (the range highlighted in yellow in fig. 8b). We obtain the offset estimate of

225 years and a 68% confidence interval from -25 to 425 years.

The difference between the ice-core offsets represents another indirect estimate of the offset between the polar timescales, which in this way is determined to be 150 years, close to the offset of 125±33 years directly obtained by synchronizing the G2B event in the $^{10}Be$ fluxes.

Our result for GICC05 is smaller than the 550-year offset obtained by Adolphi et al. (2018) but fully

consistent within the uncertainties of their estimate (95% probability interval: 215 – 670 years). For the H82 dataset, our analysis leads to offsets of about $500 \pm 200$ and $220 \pm 500$ years, for GICC05 and WD2014, respectively, which more closely reproduces the Adolphi et al. (2018) finding.



In the following, we will assume the U/Th ages to be correct and shift the ice-core time scales accordingly, although, part of the offset could also be attributed to U/Th being too old because of
measurement uncertainties or age modelling issues (Corrick et al., 2020).

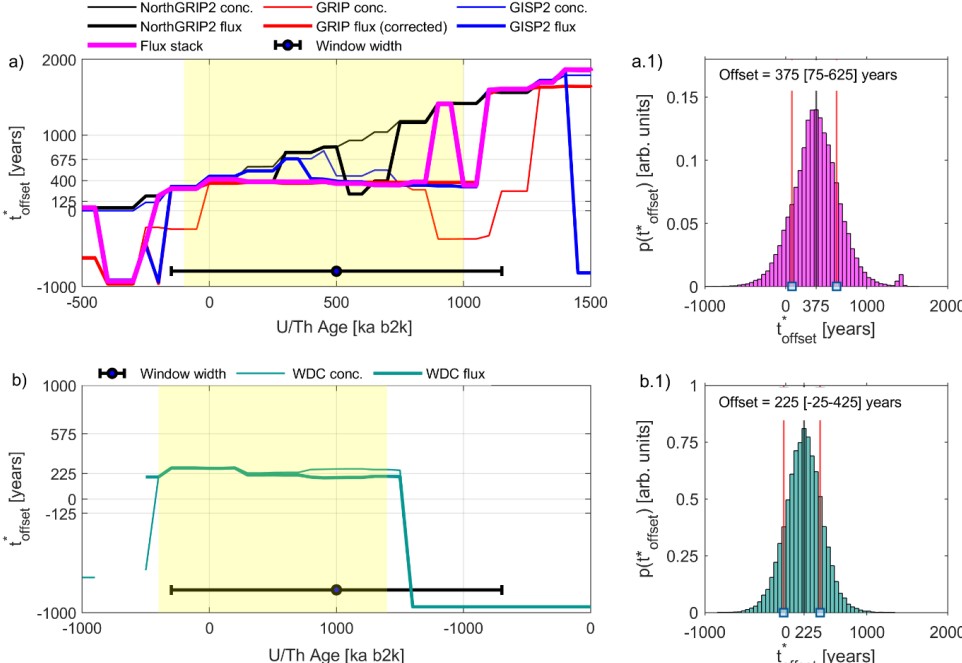

*Figure 8 Wiggle-matching result around the G2B event for Greenland (a) and the WDC core (b). The Matlab code provided in the Supplement allows to reproduce this figure by running the function 'wiggle_matching_sinnl_et_al_2022'. The time scale offset function $t^*_{offset}(\vec{W})$ of each ice-core dataset was*
*calculated as the mode of the underlying 2-dim probability density function, estimated by the algorithm (Adolphi et al., 2016). The window width (horizontal black bar) is highlighted to show that each data-point represents the data comparison within the windows $\vec{W}$. Across the intervals highlighted in yellow, the individual ice-core datasets agree about the offset for each ice-core timescale. Outside these intervals, curve instability is caused by the lack of appropriate matching features in the data. (a) The Greenland offset curve,*
*calculated after enlarging the $\Delta^{14}C$ uncertainties of the concentration-based data. (a.1) Monte-Carlo study of the Greenland offset uncertainty. Sampling each offset curve's probability envelope within the yellow interval, as outlined in the Methods, returns a histogram representing the probability of the average offset. From the histogram, we compute the 68% confidence interval of the offset (red lines). (b) For WDC, the offset curve of the concentrations and fluxes are the only available datasets for the calculation. The average offset within the*
*interval (yellow) is computed similarly as for GICC05, with the Monte-Carlo histogram shown in panel (b.2).*


## 4 Discussion

### 4.1 Climate compared after synchronization

In figure 9 we show the radionuclide and climatic data after the synchronization proposed in section 3.3.

To reconstruct the sequence of events during the LGM, we need to estimate the onset and termination of HS-2. We suggest the onset of HS-2 to be defined by the $\delta^{18}O_{calcite}$ slope in Asian speleothems (fig. 9b) (Cheng et al., 2021). Here, we identify the onset of the HS-2 signal at Hulu Cave by the same methods used for the AIM-2, combining the WDC_breakpoint function with a Monte Carlo iteration. By averaging the onset in the two $\delta^{18}O_{calcite}$ onsets at Hulu Cave, one for each dataset, we obtain the

HS-2 onset to be 24.71±0.04 ka b2k. This is earlier than, albeit compatible within 2 σ with the definition by Li et al. (2021) of an HS-2 onset at around 24.48±0.08 ka b2k, based on another speleothem (Furong, China).

*Table 3 Event onsets and terminations discussed in this work, corrected for the time-scale offsets, with 1σ confidence intervals. The events are listed from oldest to youngest (although overlap may occur because of the*
*uncertainty bounds). The Greenlandic sequence of GIs and GSs is fixed by the NorthGRIP layer count, therefore the uncertainties of these events are correlated and their order fixed, but they can be shifted as a group. In the 'Original Calendar age' column, the onset of GIs and dust terminations are based on the original definition by Rasmussen et al. (2014) with reported uncertainties related to the event definition only. The dating uncertainty is dominated by the MCE, which is about 600 years at this age. The shape of the AIM-2 and*
*$CH_4$ at WDC were assessed as described in the main text. In the column 'Age according to wiggle-matching synchronization', GICC05 has been shifted by $375^{+250}_{-300}$ years, while the WD2014 ages were shifted by $225^{+200}_{-250}$ years, both towards older ages. The uncertainty on the new age was propagated from the original uncertainty and the offset uncertainty, using the asymmetric 1σ-boundaries of the offset.*

| Event | Archive | Age according to wiggle-matching synchronization (y b2k $^{+\sigma^+}_{-\sigma^-}$) | Original Calendar age (y b2k) |
|---|---|---|---|
| HS-2 onset | Hulu $\delta^{18}O$ | 24710 ± 40 | 24710 ± 40 |
| Dust onset | NorthGRIP $Ca^{2+}$ | $24525^{+250}_{-300}$ | 24150±10 |
| AIM-2 warming onset | $\delta^{18}O$ WDC | $24497^{+200}_{-250}$ | 24272 ± 35 |
| Methane onset | WDC $CH_4$ | $24285^{+230}_{-270}$ | 24060±118 |
| AIM-2 cooling onset | $\delta^{18}O$ WDC | $23825^{+360}_{-390}$ | 23600±300 |
| Dust termination | NorthGRIP Ca | $23755^{+250}_{-300}$ | 23380±20 |
| Start of GI-2.2 | $\delta^{18}O$ Greenland | $23715^{+250}_{-300}$ | 23340±20 |
| Start of GS-2.2 | $\delta^{18}O$ Greenland | $23595^{+250}_{-300}$ | 23220±20 |
| Start of GI-2.1 | $\delta^{18}O$ Greenland | $23395^{+250}_{-300}$ | 23020±20 |
| Start of GS-2.1c | $\delta^{18}O$ Greenland | $23275^{+250}_{-300}$ | 22900±20 |



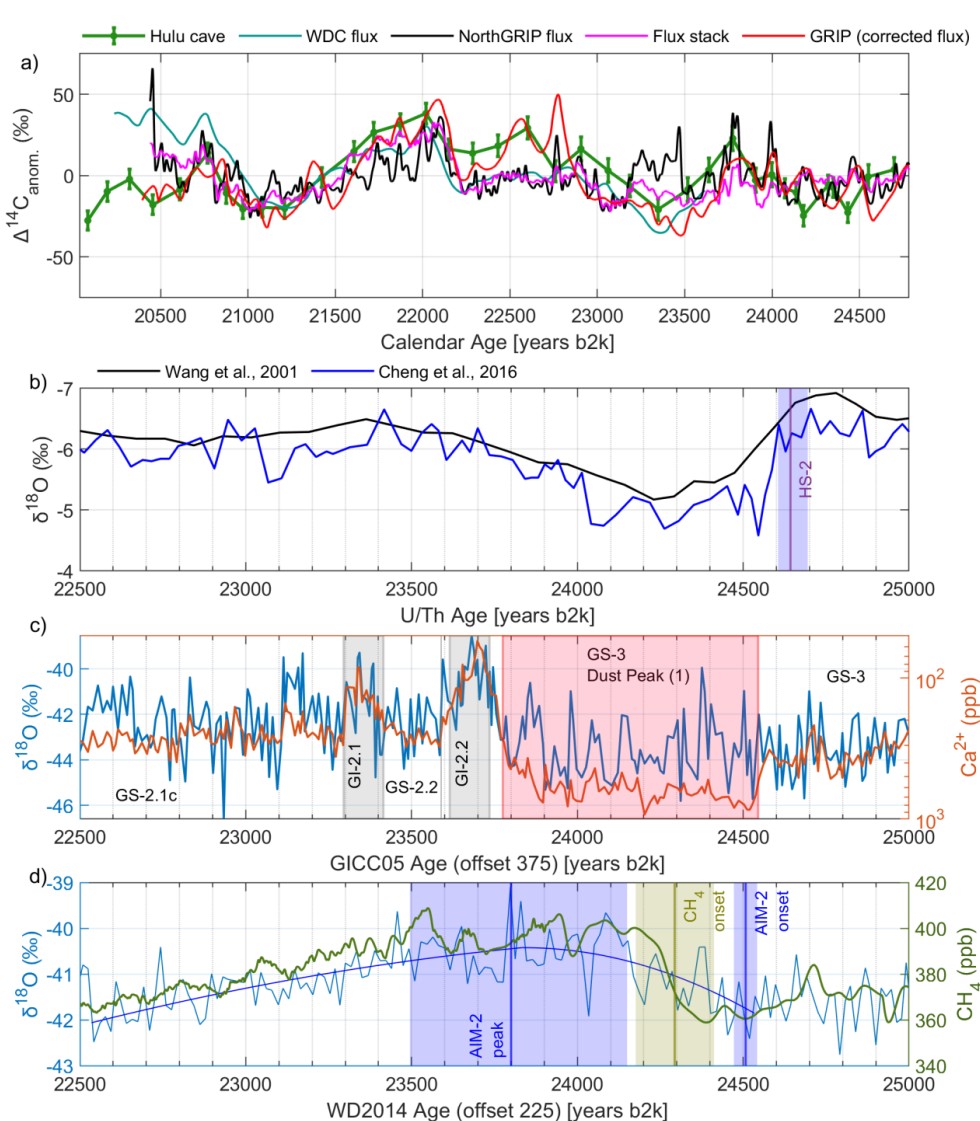


*Figure 9 Radionuclide and climatic data after the synchronization.*
*The Greenlandic records were shifted by 375 years towards older ages; the WDC records were shifted by 225 years towards older ages. (a) Radionuclide data used for wiggle matching plotted on the synchronized time scales.*

*(b) Hulu-cave isotopes (reversed y-axis) highlight the suggested onset of HS-2 (Cheng et al., 2016). A lower-resolution dataset is also plotted for comparison (Wang et al., 2001). The termination of HS-2 is not defined, as multiple possibilities arise by comparing the two datasets.*



*(c) Greenlandic proxy data on the shifted GICC05 time scale. Calcium is presented on an inverted log-axis to better compare to the Hulu record.*

*(d) The AIM-2 period aligns both with the dust peak and the HS-2 in Hulu $\delta^{18}O_{calcite}$. The onset of the methane increase occurs about 210 years after the onset of the AIM-2 warming, although we have to consider uncertainties in the gas-age reconstruction. The AIM-2 peak occurs immediately before the GI-2.2, with a wide uncertainty band related to the unclear shape of the AIM-2.*

With the time scale correction for WD2014, the onset of the AIM-2 warming occurs synchronously

with the HS-2 onset, within the uncertainties determined both by the wiggle matching and by the AIM-2 fit, although a delay would be expected given the centennial-scale response of Antarctic climate to Northern Hemisphere changes (Pedro et al., 2018; Svensson et al., 2020).

The AIM-2 maximum (i.e. the peak of $\delta^{18}O$ in Antarctica) was here defined by considering only the data from WDC. By visual inspection, the peak of the AIM-2 signal could either be identified as two

marked $\delta^{18}O$ peaks at around 23.8 ka b2k (shifted WD2014 age) or as a prolonged plateau between 23.5 and 24.2 ka b2k. We observe that the 23.8 ka b2k $\delta^{18}O$ peak in WDC occurs together with or slightly before GI-2.2, which is a similar result one obtains by merely synchronizing the data using the G2B event tie point (fig. 6). As the shape of the AIM-2 signal differs across Antarctic ice cores (Veres et al., 2013), a comparative study of other ice cores would improve this result. We cannot

provide an Antarctic comparison in this context, as the WD2014 chronology does not currently apply to other ice cores, hence an updated Antarctic synchronization across AIM-2 would be required.

The methane increase in WDC is registered ~210 years after the AIM-2 warming started and the HS-2 onset. This supports the theory by Rhodes et al. (2015) of an increased southern biogenic methane production as a delayed response to extreme northern-hemispheric cooling. The high methane levels

appear to last between 460 and more than 1000 years, depending on how one defines the end of the methane plateau.

## 4.2    Discussion on causes of the offset for GICC05

In recent work by He et al. (2021) it was suggested that, although not reflected by Greenlandic water isotope records, a shutdown of the AMOC likely occurred during Heinrich Stadial 1, which was

stronger than the AMOC shutdown of a 'regular' GS, producing a period of extreme winter cooling. They concluded that, rather than Greenland not experiencing any additional cooling during HS-1 (as proposed by Landais et al., 2018), the imprint of the cooling in the $\delta^{18}O_{ice}$ signal was cancelled by the effect of having less winter snow and by an increase of the $\delta^{18}O_{ice}$ level of summer snow, due to the first warming of the deglaciation. The flatness in the water isotope data is, in their analysis, the result





of these counteracting effects of Greenland cooling: the HS-1-specific reduction in winter
precipitation and the $^{18}$O-enriched summer precipitation.

As much as the GICC05 layers are concerned, He et al. (2021) model a possible scenario over
Greenland with a drastic decrease of winter precipitation by about 50% at the onset of HS-1. On the
other hand, they predict the summer precipitation to increase steadily as the deglaciation progresses,
which is largely unaffected by the AMOC shutdown. This could potentially have produced thin and
irregular layers at the onset of HS-1, with the summer part of the layer gradually increasing over time.
Acknowledging the 125-years offset between GICC05 and WD2014 at 22 ka b2k, and the 375 years
offset between GICC05 and the U/Th timescale, we proceed by discussing where the offset could
have originated. According to Adolphi et al. (2018), the transfer function from GICC05 to IntCal is
near to zero at 13 ka b2k, the closest tie-point younger than the G2B event. Occurring between these
two time-horizons, the HS-1 period lasted from about 18 to 14.7 ka b2k, and is most often regarded
as ending at the onset of GI-1. The authors of GICC05 retained the prior subdivision of the
corresponding stadial, GS-2.1, in 3 sections (termed a, b, and c) of which GS-2.1a starts at 17.48 ka
b2k and ends at the onset of GI-1 (Björck et al., 1998; Rasmussen et al., 2014). The onset of GS-2.1a
was originally defined by a 2$^{nd}$-order water isotope dip but occurs synchronously with an increase in
the Greenland ice-core calcium profiles at roughly 17.6 – 17.4 ka b2k.

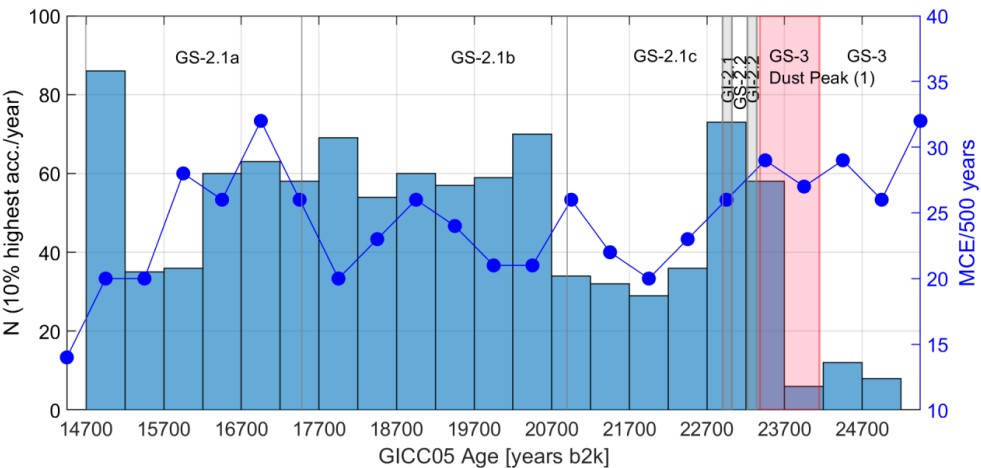

*Figure 10 Highest accumulation years (histogram) and MCE (dark blue curve) based on the GICC05 timescale, computed on the same 500-years intervals.*



*The MCE appears rather constant over GS-2, except for some higher values at ~16.7 ka b2k. On the other hand, the thickest layers (>0.09 m/year; corresponding to the 10% thickest layers in the entire period) are located preferentially across GS-2.1b and 2.1a, with a sharp onset at 20.7 ka b2k. The first bin (14.7-15.2 ka b2k) also contains relatively thick layers, possibly an effect of the onset of GI-1. As expected, the thickest layers are often found in the brief interstadials, while over the dust peak the frequency of high-accumulation years is much lower. The thickest layers across GS-2.1b and 2.1a have as many high-accumulation years as interstadial periods, which is surprising given that this period was very cold and had very low annual precipitation (Kindler et al., 2014).*



If GICC05 missed a large number of annual layers across HS-1, the age of the onset of GS-2.1a would move towards older ages by a similar amount. We speculate that the onset of GS-2.1a could correspond to the onset of HE-1 (and thus the HS-1 period) and that calcium could be used as a signature of the change of atmospheric circulation influencing dust transport to Greenland during HS-1 as well as HS-2.


Part of the under-counted layers could have contributed to the MCE. The MCE is derived from the number of features that could be annual layers but are not clearly identified as such; it does not properly account for layers not resolved by the data. Within GS-2.1a, the MCE grows by ~150 years, meaning that 300 uncertain layers were counted as 0.5±0.5 years. If we assume that all the uncertain annual layers in GS-2.1a were annual layers, an additional 150 years would appear in the chronology, and the MCE would decrease by 150 years. As we have seen, according to He et al. (2021), a large number of very thin annual layers is a plausible scenario at least in the first part of HS-1 until the increasing summer precipitation alleviated the problem. As these layers are possibly unresolvable by the data, they would not be counted as uncertain layers, hence the MCE is probably not faithfully representing the full GICC05 uncertainty in sections where layer thicknesses are very small compared to the resolution of the data.



In addition to mis-assigned uncertain layers and layers missed altogether due to marginal data resolution, we here propose another explanation of what could have caused the under-count of GICC05 layers. The layers that are on the higher end of the thickness distribution, even if they were classified as 'certain', might indicate where very low or absent winter precipitation could have made multiple annual layers appear as one.


We highlight the 10% years with highest accumulation in fig. 10, where we also show the MCE, both per 500-year interval. The accumulation threshold is set at 0.09 m/year by integrating the highest 10% of the empirical distribution of the accumulation between 14.7 and 25 ka b2k. In the accumulation histogram of fig. 10, values are consistently high over the entire GS-2.1b, with high-accumulation years occurring as frequent as in the short interstadials. For the MCE, it appears that observers of




GICC05 encountered more issues at the onset of GS-2.1a, whereas the MCE stayed constant over the
rest of GS-2.1. Hence, we suggest that a dating bias could have accumulated across GS-2.1a and GS-
2.1b due to increased difficulty in identifying annual layers caused by a weakening of the annual
signal due to reduced amounts of winter snow. However, we find no clear evidence that this
phenomenon occurs at the onset at HS-1 (17.7 ka b2k).

## 5    Conclusion

In this study, we have presented two new [10]Be datasets that allow both a bipolar synchronization and
a comparison of polar climate to the Hulu speleothem archive during the 20-25 ka b2k period. The
new NorthGRIP [10]Be dataset suggests a dating offset between GICC05 and U/Th time scales of
$375^{+250}_{-300}$ years, which is less than the 550-year estimate by Adolphi et al. (2018), albeit consistent
within uncertainties. Likewise, an offset was found between U/Th Hulu Cave dates and the WD2014
time scale of $225^{+200}_{-250}$ years, which is supported by the visual alignment of the datasets.
For the WD2014 time scale, the offset can be compared to the uncertainties quoted for the time scale
by Sigl et al. (2016). Between 20 and 25 ka b2k, 1σ uncertainties are between 100 and 125 years,
which are smaller than the offset we find. This suggests that the authors of WD2014 may have
underestimated their uncertainties, similar to GICC05, and that the layer counting in the glacial was
also more challenging than acknowledged at first, but the two estimates are nonetheless within 2σ.
However, the gradual warming observed between 19 and 15 ka b2k (Pedro et al., 2011) likely did not
lead to fast changes in the annual layers' expression. The WD2014 issues may thus be mostly related
to the low resolution of the data used for counting between 15 and 26 ka b2k (Sigl et al., 2016), which
may have limited the accuracy of the time scale.
In terms of the sequence of events between 20 and 25 ka b2k, our time-scale offset does not change
their order. However, the onset of the Greenlandic dust peak moved to be roughly synchronous with
the signal in the Hulu speleothem that has been linked to the HS-2 onset. Observing the onset of AIM-
2 occurring relatively near to the dust and Hulu signals, we support the hypothesis that the AIM-2
warming is related to an AMOC shutdown during HS-2. The $CH_4$ increase onset in Antarctica also
moved closer to the HS-2 onset, supporting the theory by Rhodes et al. (2015) of a long overlap of
the methane plateau with the HS periods. The termination of the Greenlandic dust peak and the almost
synchronous onset of GI-2.2 were brought closer to the AIM-2 peak. We observe that the AIM-2
maximum, at least within WDC δ[18]O data, occurs together or before the GI-2.2 onset. This may





indicate an exception to the average delay of 122±24 years found within other GI-AIM pairs, where

the GI generally occurs before the AIM breakpoint (Svensson et al., 2020). Another piece of evidence

suggests a deviation from the normal bipolar seesaw mechanism: It seems likely that the very brief

GIs are not the primary cause of the sustained Antarctic gradual cooling taking place during the first

millennia of the GS-2.1, but could be an indication that the AMOC resumed or gained more strength

after GI-2.1 and into GS-2.1.

We investigated several possible reasons why the annual-layer counters of GICC05 could have

missed annual layers across GS-2.1. We find that the MCE of GICC05 alone cannot account for all

missing layers in the GS-2.1, but that either very thin annual layers or missing winter precipitation

could have made it difficult, if not impossible, to identify the thinnest layers with the available data,

and thus made the MCE estimation less robust.

## 6   Acknowledgement

G.S. and S.O.R. acknowledge support via the ChronoClimate project funded by the Carlsberg
Foundation.

F.A. acknowledges support through the Helmholtz Association (grant VH-NG-1501). R.M.
acknowledges support from the Swedish research council (grants DNR2013-8421 and DNR2018-
770     05469).

K.W., T.W. and M.C. were supported by NSF grants 0839042 and 0839137. These authors
acknowledge the support of the WAIS Divide Science Coordination Office at the Desert Research
Institute in Reno, Nevada, for the collection and distribution of the WAIS Divide ice core (Kendrick
Taylor, NSF Grants 0440817 and 0230396), support from the NSF Office of Polar Programs, which
funds the Ice Drilling Program; Raytheon Polar Services for logistics support in Antarctica; and the
109th New York Air National Guard for transport of equipment, people and ice cores in Antarctica.
Finally, we acknowledge Geoff Hargreaves and his staff at the NSF Ice Core Facility, for archiving
of the ice core and organizing and hosting ice core processing.

NGRIP was directed and organized by the Department of Geophysics at the Niels Bohr Institute for
Astronomy, Physics and Geophysics, University of Copenhagen. It was supported by funding
agencies in Denmark (SNF), Belgium (FNRS-CFB), France (IPEV and INSU/CNRS), Germany
(AWI), Iceland (RannIs), Japan (MEXT), Sweden (SPRS), Switzerland (SNF) and the USA (NSF,
Office of Polar Programs).



## 7 Data and Supplement

$^{10}$Be data of NorthGRIP and WDC are made available in the excel documents provided with the Supplement of this paper. Moreover, Matlab codes for reproducing fig. 8 and fig. 9a are provided in the Supplement, as well as the supplementary figures.

## 8 Methods Appendix

### 8.1 WDC measurements

Ice samples from the WAIS Divide ice core were stored and processed at the NSF Ice Core Facility in Lakewood, Colorado. Of each 1-m section from 2453-2599 m depth, a thin slice (DD-16) with a cross-section of ~2 cm$^2$ from the outside of the core was cut for $^{10}$Be analysis and shipped in frozen form to Purdue University. For each $^{10}$Be sample, two consecutive core sections were combined,
yielding a total mass of 350-400 g per sample. Ice samples were weighed, melted, and acidified with a solution containing ~0.18 mg of Be carrier. The samples were passed through a 30 µm Millipore filter and loaded onto a 3 ml cation exchange column (Dowex 50WX8) from which the Be fraction was eluted and purified following established procedures (Finkel and Nishiizumi 1997; Woodruff et al. 2013). The Be fraction was precipitated as Be(OH)$_2$, transferred to a small quartz
vial and heated in a tube furnace at 850 ℃. The BeO was mixed with Nb powder (Alfa Aesar, -325 mesh, Puratronic, 99.99%) and pressed into a stainless-steel cathode. The $^{10}$Be/$^9$Be ratios of samples and blanks were measured by accelerator mass spectrometry at Purdue's PRIME laboratory (Sharma et al. 2000), relative to well-documented $^{10}$Be/Be AMS standards (Nishiizumi et al. 2007). The measured $^{10}$Be/Be ratios of the samples were corrected for an average blank $^{10}$Be/$^9$Be ratio of
$(10 \pm 3)$ x 10$^{-15}$ which corresponds to typical blank corrections of 1-2% of the measured values. The blank-corrected $^{10}$Be/$^9$Be ratios, combined with the sample mass and amount of Be carrier added yield $^{10}$Be concentrations ranging from 3.1 to 5.4 x 10$^4$ atoms/g (fig. 1d). Typical uncertainties (1σ) in the measured $^{10}$Be concentrations are 1.5-3.5%. The measured $^{10}$Be concentrations were not corrected for radioactive decay, which would increase all values by 1.0-1.2%.



## 8.2 NorthGRIP measurements

### 8.2.1 Ice cutting

At present, the required minimum weight for each $^{10}$Be measurement is around 120 g. About half of the samples, in alternate order, had previously been cut in the shape sticks for gas measurements (section area of 3.5x3.5 cm$^2$), with a weight of around 600 g per bag (55 cm). The "gas sticks" were thus cut into 4 parts, resulting in pieces of 13.75 cm, corresponding to a resolution of about 7.5 years. The other half of the samples were cut from the archive piece to have a section area of ~2x3 cm$^2$. Each bag of these was then cut into two parts, hence each of such pieces weighs around 180 g and corresponds to about 14 years resolution. The campaign was performed with the necessity of keeping the total number of samples at around 320 for cost reasons. Since the total number of cut ice samples was 470, the pieces were selected to alternate between adjacent samples at 7.5 years resolution, to minimize the age gaps; the resulting $^{10}$Be data consequently shows some data gaps.

### 8.2.2 Sample preparation and AMS Measurement (ETHZ)

Ice samples of 150-180 g were weighed and a solution containing 0.15 mg Be carrier ($^9$Be) was added, as well as 1 mg of Cl carrier. The samples were melted, not filtered, and run through cation exchange columns, from which the Be fraction was extracted. The melt water was further run through anion exchange columns to retain the chlorine content and stored for future $^{36}$Cl measurement. The resulting Be(OH)$_2$ was heated in steps to 850° to obtain BeO and mixed with Niobium powder. Five blank samples of Milli-Q water were also measured for background assessment. More details about the most recent preparation protocol which is used at the Lund Laboratory can be found in Nguyen et al. (2021).

The $^{10}$Be/$^9$Be ratios of the samples were measured in July 2020 at the AMS facilities at ETH in Zurich (Christl et al., 2013), where a total of 322 measurements were performed. Measured $^{10}$Be/$^9$Be ratios were normalized to the ETH Zurich in house standards S2007N and S2010N which in turn have been calibrated relative to primary standards provided by K. Nishiizumi (2007). An average blank correction of $^{10}$Be/$^9$Be = 1.3 ± 0.3 10$^{-14}$ was applied to correct for $^{10}$Be introduced with the Be-carrier. The blank correction corresponds to 2-3% of the measured $^{10}$Be ratios. Calculated $^{10}$Be concentrations range between 1.8 and 8.7 10$^4$ atoms/g and were decay corrected using a half-life of 1.387 ± 0.016 10$^6$ years, which, at these ages, produces a signal increase of only about 1%. The final uncertainties of the $^{10}$Be concentrations were between 1 and 10 %.



### 8.3 From concentrations to fluxes

The concentration to flux conversion, for ice cores, is obtained by calculating $\varphi = \rho_{ice}\gamma\alpha$, where $\varphi$ is the flux, $\gamma$ is the $^{10}$Be concentration, $\alpha$ is the accumulation rate, and $\rho_{ice}$ is the ice density (0.917 g/cm$^3$).

### 8.4 Wiggle-matching algorithm settings

Thanks to the higher resolution of ice-core data with respect to Hulu data and thanks to the carbon-cycle model producing a continuous output, we approximate the ice-core modelled $\Delta^{14}$C input as a continuous function of time, i.e. we resample it annually by linear interpolation. The second input, the Hulu-cave data, is a collection of discrete data points. The $\Delta^{14}$C uncertainties are about 6‰ for the Hulu Cave data and 5‰ for the ice-core data. The datasets are detrended within each observation window before computing the probabilities. In particular, the ice-core $\Delta^{14}$C is first shifted according to the scanned time offset, then detrended with respect to the observation window, and resampled to the Hulu-cave sampling points.

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
