# Peer review of "Synchronizing ice-core and U/Th time scales in the Last Glacial Maximum using Hulu Cave 14C and new 10Be measurements from Greenland and Antarctica"

_Climate of the Past, 2022_

## Author Comment (AC1)

**Reply to Anonymous Referee #1 regarding cp-2022-62**

"Synchronizing ice-core and U/Th time scales in the Last Glacial
Maximum using Hulu Cave 14C and new 10Be measurements from Greenland and
Antarctica" by Giulia Sinnl et al., Clim. Past Discuss.,
https://doi.org/10.5194/cp-2022-62-RC1, 2022

We thank An.Ref. #1 for the feedback. We have marked our responses to each comment in blue.

>>In this study, new [10]Be NorthGRIP and WAIS Divide ice core measurements were compared to Hulu cave [14]C measurements to constrain the age scales of these records through the Last Glacial Maximum (LGM). This exercise is important for improving the understanding centennial-scale climatic events in the LGM. Radionuclide particle production changes as a result of solar activity variability. Once changes in its transport to and deposition at ice core sites are accounted for, radionuclide particle variability in proxy records is therefore independent from changes in climate. It is beneficial to use radionuclide particle records to constrain proxy age scales due to this independence. In this study, a time period characteristically similar to the Maunder Minimum identified in the LGM 10Be records was used to synchronize the Greenland (GICC05) and Antarctic (WD2014) ice core ages scales. Using this analysis, an ~125-year difference between the age scales prior to synchronization was determined. A wiggle-matching algorithm was also used to synchronize the ice core age scales to the Hulu Cave age scale. The offsets between the Hulu Cave age scale and the GICC05 and WD2014 age scales were ~375 years and ~225 years, respectively.
This study is important to publish because highly temporally resolved paleoclimate datasets can only be compared to other archives if the age scales of the datasets are accurate. An improvement in the accuracy of proxy age scales therefore leads to better understanding of the timing and progression of climate events and therefore a better understanding of the climate system.

Reply: We thank the referee for this review and for the overall encouraging words about our study. We will improve the manuscript as suggested.

>>Major comments:

>> The introduction needs to be revised. The introduction is a long description of several well-known past climate events without giving the readers any context for why they are being presented with this information. Is there something about these climate events that is unresolved that is addressed in this study? It is a nice literature review, but why is it given?

Reply: We think that the inter-disciplinarity of our study (e.g. ice cores meeting speleothems, carbon-cycle models meeting polar measurements) requires a general and complete description of the background, without giving too lengthy of an introduction. In the revised manuscript, we have improved the structure of the paragraphs, strengthening our motivations about unresolved timescale issues being problematic for climatic interpretations, while also recalling all the necessary elements that are founding for our analysis.

>>Along the same lines, in the introduction, the authors state that the "objective" of the study is a "comparison of three timescales." This isn't really an objective. The comparison is really the approach used to address the objective, which I believe is to improve the accuracy of the timing of climate events in the LGM, which is necessary to understand (eventually) the mechanisms behind them.

Reply: Thank you for pointing this out, we have changed the statement about our objectives accordingly. The time scale comparison is at the centre of our methodology, but we recognize that the broad scope of our efforts is more accurately explained as the deeper understanding of the LGM climate through an assessment of the event sequences that characterized this period. In particular, testing the LGM tie point found by Adolphi et al. (2018) has been an important objective as well, as the implied layer counting biases around the LGM were rather unexpected in light of a better agreement of the time scales, documented in the cited literature, both before and after the LGM.

>>The age scale development of the three proxies is then nicely summarized in the introduction, but again, the readers are not given any information about how the current study fits into any of it until ~line 155. It would be very helpful if the authors explained the flaws in the previous dating methods much earlier. Otherwise, the reader does not know why they are being given the summary. It needs to be made very clear that the benefit of using radionuclide records is that their variability is independent from (at least in the case of flux) climatic events. Therefore, the circular nature of dating proxies using climate events and then, in turn, interpreting the timing of those events, is avoided.

Reply: We have altered the structure to frontload our study's position and relevance in relation to the cited references.

>>The uncertainties in the age scale offsets are rather large given the small magnitude of the offsets. In the conclusion, it would be helpful if the authors could suggest ways in which these uncertainties could be reduced in future studies.

Reply: Thank you for this point, in the revised manuscript we address this issue by the following arguments. We remark that the offset between bipolar $^{10}$Be datasets (125±40 years) is the type of precise estimate that the carbon-modelled $\Delta^{14}$C cannot compete with. In the future, uncertainties in the bipolar match could be reduced even more, for example, via a more accurate volcanic matching, supported by our radionuclide matching. With the guidance of an established radionuclide match, albeit with wide uncertainties, a bipolar volcanic match in the LGM will be the most precise way to link the two ice sheets.
The issue of large uncertainties in the wiggle matching lies in resolution problems of the $^{14}$C data, in the low signal-to-noise ratio of both $^{14}$C and $^{10}$Be, and in the carbon-cycle model likely not capturing all aspects of the LGM carbon cycle. The combination of these factors leads to multiple possible fits between the two datasets.

A way to improve the matching to the speleothems would be either a higher resolution absolutely-dated tree ring [14]C data (with more reliable signal structures that we can match), or similarly resolved [14]C but from another speleothem with lower and more stable DCF than Hulu. In addition, high-resolution [10]Be records from other ice cores would reduce the uncertainties by limiting the (local) weather/climate noise in the NGRIP records.

>>The focus of this study is the LGM, but as the authors state that the age scales were not stretched in this study. How would the age scales before and after the LGM be affected by the offsets suggested here? Are offsets of a few hundred years too small to make much of a difference?

Reply: Thank you for this comment. In our analysis, at the common tie point in our focus, the G2B event, GICC05 is ~400 years younger than U/Th and 125 younger than WD2014. In the discussion paragraph 4.2, we argue that the missing layers of GICC05 where likely accumulated during GS 2.1a and 2.2b, i.e. between 15 and 20 ka b2k. The offset is likely at its maximum around 21 ka. Another tie point was found at 31 ka, where GICC05 and U/Th are only 150 years offset (Turney et al., 2016). Therefore, it appears the layers are over-counted somewhere before the LGM, otherwise the 400-year offset wouldn't reduce itself to 150 years, a point also made by Adolphi et al. (2018). In addition, Corrick et al. (2020) do not find large offsets in MIS3, supporting that the difference becomes smaller below the G2B event.

In the revised manuscript, in light of the recent publication by Dong et al. (2022), we also discuss a bipolar volcanic tie-point triplet, which is found at 24669 years b2k in GICC05 and 24589 years b2k in WD2014 (Svensson et al., 2020). Here, the event is 80 years older in GICC05 than in WD2014, i.e., the offset has opposite sign than at the G2B event. It therefore appears that more layers were counted in GICC05 over the interval between G2B and the triplet.

However, a uniform stretching GICC05 with respect to WD2014 is not advised: since the thinning of WDC layers is steeper than in Greenland, we cannot assume the offset would be evenly distributed. A further assessment of the time scale accuracy, via for example more high resolution [10]Be measurements in WDC across AIM-2, would be required.

>>The conclusion made in this study is that age scale corrections need to be made to the ice core records, and that the problem is the result of annual-layer-undercounting. Why does the problem lie with the ice cores? Is the age scale of the Hulu Cave record that much more certain?

Reply: The measured timescale offsets are relative between timescales. It is more obvious to us to investigate the issue in ice core layers, as they pose clear challenges in terms of identification, e.g. because layers are very thin during the LGM. From the published U/Th dating uncertainties we cannot conclude that U/Th could carry any absolute error, and we are not in a position to question these uncertainties. We are not aware of estimates in the LGM period showing Hulu being dated too old, but Corrick et al. (2020) do mention a possible issue of sub-optimal sample positioning of U/Th and/or $\delta^{18}O$. If one uses the argument of $\delta^{18}O$ synchroneity across Asian speleothems, considering all caveats because of climate effects, then one can observe a spread of the HE-2 onset between Hulu and the Cherrapunji speleothem (Dong et al., 2022) of about 100 years, where the Hulu record shows the oldest onset of HE2. This could indicate that dating issues in

Hulu are in fact present (given that Cherrapunji was very carefully counted over the LGM using annual lamina). If Hulu was 100 years too old in the LGM, then the offset to GICC05 and WDC would obviously be smaller. However, there is no radionuclide data for the Cherrapunji speleothem, so our methodology remains strongly dependent on the Hulu $^{14}$C record. We have presented these arguments in the revised manuscript.

>>Minor comments:

>>Line 60: This sentence is confusing: "During this time, a phase of massive discharge of icebergs from the Laurentide ice sheet was inferred from the ice-rafted debris content of North Atlantic marine sediments, defining the occurrence of the Heinrich Event 2 (HE-2; Bard et al., 2000; Peck et al., 2006)." You mean that HS2 happened at the same time as the LGM, right? Simplify this sentence.

Reply: We have clarified the sentence, thank you.

>>Lines 63-66: "The term Heinrich Stadial (HS) is often used to indicate the period affected by the HE. The duration of HS-1, for example, is limited to the 14.5-17.5 ka b2k interval within GS-2.1 (Broecker and Barker, 2007), while for HS-2, a correspondence with the late 65 GS-3 is often argued for, based on speleothem water isotope records (e.g. Li et al., 2021)". It would be very useful if the timing of each Heinrich Stadial and each Greenland Stadial referenced was defined and easily referenced. Maybe a table could be added?

Reply: Thank you, we have added a table with the definitions of GS1, HS1, HE1, GS2, HS2, and HE2 to precisely refer to the nomenclature adopted in this work.

>>Even though it is commonly used, please add a sentence defining the IntCal20 curve.

Reply: We have added an explicatory sentence.

>>Line 100: "and GICC05 was extended to these ice cores." This is odd phrasing. I'm not sure what this means.

Reply: We have edited this sentence to clarify better. We mean that GICC05 was transferred from NGRIP to the other Greenland ice cores via volcanic matching.

>>Why was before 2000 (b2k) used instead of the conventional, before 1950?

Reply: That is because for Greenland time scales the convention b2k is commonly used and is endorsed for other ice core time scales as well. We will add a conversion to BP throughout the paper, where most relevant.

>>Lines 117-119: "The authors duly excluded the GI-2–AIM-2 pair from their lead-lag analysis, firstly because the GISP2 CH4 record did not support synchronicity with the GI-2

temperature increase, and, secondly, because the older HE-4 and HE-5 were similarly associated with higher CH4 levels." What is meant by "because the older HE-4 and HE-5 were similarly associated with higher CH4 levels?" Does this mean that GI2 and the HE's can't be distinguished, and that the HE's are associated with stadials?

Reply: Thank you, we have rephrased this sentence in the revised manuscript. We intended to recall that HE-4 and 5 are also recorded in the methane data, which suggests that the methane increase during GS-3 is associated with HE-2, rather than with GI-2.

In the Matlab code provided as supplement to WAIS Project Members (2015), it is stated that the $CH_4$ rise in GS-3 is "not coincident with the Greenland temperature rise, as is clear from e.g. GISP2 d15N and $CH_4$ data. This CH4 rise is likely to reflect southern sourced CH4 during the Heinrich stadial, as is also observed during HS4 and HS5; i.e. DO 8 and DO 12 are preceded by increased CH4 concentrations also." Therefore, the WAIS synchronization excluded the AIM-2-GI-2 pair, considering the interpretation issues of the $CH_4$ signal.

In the Antarctic $CH_4$ data, within GS 9 and 13, there are double-level signals where the first level is thought to be associated with HE4 and HE5, while the second and higher level can be aligned with the following onset of GIs in the Greenland record. However, in the case of GS-3, this double-level methane structure is less clear and the matching with GI-2 is not possible. The $CH_4$ shapes of HS4, HS5, and HS2 are compared in Rhodes et al. (2015) at fig. 2.

>>Lines 128-130: This needs to be more prominent: "Resolving some time-scale issues, which we will delineate shortly, will clarify the distinctive timing factors of the global climate around HE-2, compared to the 'conventional' bipolar seesaw scenario."

Reply: In the restructuring of the introduction, we have given more importance to this point.

>>Lines 31-33: This point should also be more prominent: "Traces of volcanic eruptions and cosmogenic radionuclides provide synchronization tools that do not rely on the precise identification of climatic match-points and on the assumption of their synchronicity."
It is hard to see the effect of the 10Be flux calculation when the concentration and flux aren't plotted together.

Reply: In the restructuring of the introduction, we have given more importance to this point.

>>Lines 241-243: "A carbon-cycle model (here the box-diffusion model by Siegenthaler, 1983) is necessary to derive the atmospheric Δ14C signal, i.e. the decay and fractionation-corrected ratio of 14C/12C relative to a standard (Stuiver & Pollach, 1977), from the measured ice-core 10Be." Please clarify what is meant by "from the measured ice core 10Be." How was the 10Be used in the model?

Reply: We have clarified this in the revised manuscript. The $^{10}$Be is normalized, amplified by 20%, and provided as an input signal to the model. This is done under the assumption that the variations of $^{10}$Be, measured in ice cores, can be converted to a global $^{10}$Be production rate (after correcting for a possible polar bias). The global variations of $^{10}$Be are theoretically proportional to the $^{14}$C global production rate variations, which is what is fed into the model.

>>Lines 258-260: "The strength of the geomagnetic field directly affects both the 10Be and 14C production rates. Although each radionuclide may be affected differently (Masarik & Beer, 2009), most studies do not find any significant difference in production rates (e.g. Kovaltsov et al., 2012; Herbst et al. 2017)."… I thought that the geomagnetic field did affect production rates? Please clarify this sentence.

Reply: We realize the two sentences might seem contradictory and we have made a clarification. We mean that recent studies suggest that the geomagnetic field affects the two radionuclides by the same proportion (i.e. a 20% change in the global $^{10}$Be production rate implies also a 20% change in the global $^{14}$C production rate), while only Masarik and Beer (2009), to our knowledge, find that a ratio of 1.3 is to be expected between $^{10}$Be and $^{14}$C production rates because of the geomagnetic field (a 10% change in $^{10}$Be would go together with a 13% change in $^{14}$C).

>>Lines 256-258: "To compare the measured and the modelled Δ14C, in this study we will make use of linear detrending, as this largely removes the systematic offsets associated with the unknown carbon cycle history and inventories." Were both datasets detrended? Please clarify what was done to detrend the data and which datasets were used.

Reply: We have now added an explanation that all datasets were detrended in the same way by, first, selecting the data in a consistent timeframe for all datasets (20-25 ka b2k) and, then, using the Matlab function detrend(), which subtracts the best straight-fit line from the data.

>>What is the orange in Fig. 5?

Reply: In fig. 5 we have added that the orange indicates the intervals of missing data (like in fig. 1a)

>>Line 505-506: "The stack is shown in fig. S1, with uncertainty bands derived from the standard deviation of the 1000 simulated fluxes." Is there a reason the stack isn't shown in the main manuscript? Is it not particularly relevant?

Reply: The stack is later used to apply the wiggle-matching as if it constituted an additional, independent, 'ice core dataset'. Because of the stacking method adopted, the wiggle-matching result of the stack is slightly different than simply averaging the output from the individual ice cores. We considered the stack itself not to be informative enough to dedicate a full figure in the main text, but, given your comment, we interpret that the reader might expect to see the stack directly where it is mentioned. We have added it in a panel in fig. 6 and used it to determine its own bipolar offset to Antarctica.

>>Lines 516-518: "The G2B event: a relatively abrupt increase of 30 ‰ in the modelled Δ14C from 10Be, reaching its maximum at 21,725 years b2k (GICC05 ages), about 100 years after the maximum is reached in 10Be fluxes." This is a bit confusing because this event is called the "G2B" event, but then it is stated that happens 100 years after the 10Be. Please explain. Also, if the timing were the same as the 10Be event, wouldn't you expect this, considering that the 10Be data are an input that is used to produce the

modelled Δ14C data?

Reply: There is a dampening and a delay between changes in $^{14}C$ production rate (reflected in our $^{10}Be$ measurements) and in the atmospheric $^{14}C$ concentration (measured as $\Delta^{14}C$ ) due to the carbon cycle (due to the large active $^{14}C$ reservoirs) which needs to be accounted for when determining the true age of the solar minimum. Hence, the $^{10}Be$-signal, which is rapidly deposited, leads the manifestation of the G2B signal in $\Delta^{14}C$. We have clarified the two ages accordingly in the text, making it clear that the event is detected in $^{10}Be$ and $^{14}C$ but for the synchronisation the about 100 years carbon cycle-related delay needs to be accounted for.

>>Table 3 is very helpful!

Reply: Thank you.

>>Lines 654-656: "We cannot provide an Antarctic comparison in this context, as the WD2014 chronology does not currently apply to other ice cores, hence an updated Antarctic synchronization across AIM-2 would be required." I don't quite understand this. Isn't WD2014 applied to the South Pole Ice core (SPICE) and to Skytrain (paper recently submitted to CP)?

Reply: We added the SPICE ice core to the Antarctic picture. We attach a preliminary figure for the revised manuscript, showing the gas data and $\delta^{18}O$ data of the WDC and Spice ice cores on their common time scale, which shows the agreement of the two ice cores regarding the shape of AIM2 and the associated $CH_4$ signal. The data show a broad agreement in the shape of the AIM-2 and of the $CH_4$increase. The Skytrain data is still under review and subject to revision, however the methane shown at fig. 5 of Mulvaney et al. (2022, in revision) appears to agree with WDC in the LGM. We have added a comment in the main text remarking the average features of the Antarctic signals.

[Figure]

>>Technical changes:

Reply: All the minor technical changes have been considered in the revision. We only add replies to the more complex comments.

>>Line 50: change "being debated" to "under debate"

>>Line 52: add comma after "cold period"

>>Line 54: "was established to have lasted until"…change to something like "suggest that the LGM lasted until…"

>>Line 56: "since it coincides with the age limits of our new Greenland 10Be dataset"…what does this mean?

Reply: We mean that 20-25 ka b2k was chosen as the measurement time frame for NGRIP, which is chosen as axes limits in all our figures. Since this is in broad correspondence with the LGM, which does not currently have a precise global stratigraphic definition, an informal correspondence between LGM and "measurement timeframe" is often adopted in this work.

>>Line 108: Define $\delta^{18}O_{ice}$

Reply: We have added a definition.

>>Figure 1 caption: The second "d" should be "e"
>>Sections 2.1 and 2.2: references to the 10Be methods???

Reply: We have added the relevant citations.

>>Line 323: "quantify the impact of 10Be measurements uncertainty" change to "quantify the impact of 10Be measurement uncertainty"

>>Line 275 vs. line 390: 21.7 ka event in one line and 22.7 ka event in another. Are these different events?

Reply: No, it should be 21.7 ka event throughout. Thank you for noting this.

>>Line 451: "some eruptions are better visible in the"…change to "more visible"

>>Line 461-462: "and we obtained a timescale correction, which we apply in the following to the GRIP data." This is confusing. Please simplify this sentence.

>>Line 560: "Measured Hulu Cave used for synchronization" You should add that this is Δ14C.

>>Line 672: change "As much as the GICC05 layers are concerned," to "As far as the GICC05 layers are concerned"

>>Line 677: change "Acknowledging the 125-years offset" to "Acknowledging the 125-year offset"

>>Line 677-678: change "the 375 years offset between GICC05" to "the 375-year offset between GICC05"

>>Lines 746-747: "However, the onset of the Greenlandic dust peak moved to be roughly synchronous with the signal in the Hulu speleothem that has been linked to the HS-2 onset." Turn this into a sentence.

Kind regards,
Giulia Sinnl et al.

References:
Adolphi, F., Bronk Ramsey, C., Erhardt, T., Edwards, R. L., Cheng, H., Turney, C. S., ... & Muscheler, R. Connecting the Greenland ice-core and U∕Th timescales via cosmogenic radionuclides: testing the synchroneity of Dansgaard–Oeschger events. Climate of the Past, 14(11), 1755-1781. https://doi.org/10.5194/cp-14-1755-2018 , 2018

Corrick, E. C., Drysdale, R. N., Hellstrom, J. C., Capron, E., Rasmussen, S. O., Zhang, X., ... & Wolff, E. Synchronous timing of abrupt climate changes during the last glacial period. Science, 369(6506), 963-969. DOI: 10.1126/science.aay5538 ,  2020

Dong, X., Kathayat, G., Rasmussen, S. O., Svensson, A., Severinghaus, J. P., Li, H., ... & Cheng, H. (2022). Coupled atmosphere-ice-ocean dynamics during Heinrich Stadial 2. *Nature communications*, *13*(1), 1-14.

Mulvaney, R., Wolff, E. W., Grieman, M., Hoffmann, H., Humby, J., Nehrbass-Ahles, C., ... & Prié, F. (2022). The ST22 chronology for the Skytrain Ice Rise ice core–part 2: an age model to the last interglacial and disturbed deep stratigraphy. *Climate of the Past Discussions*, 1-30.

Rhodes, R. H., Brook, E. J., Chiang, J. C., Blunier, T., Maselli, O. J., McConnell, J. R., ... & Severinghaus, J. P. Enhanced tropical methane production in response to iceberg discharge in the North Atlantic. Science, 348(6238), 1016-1019. DOI: 10.1126/science.1262005 ,  2015

Turney, C. S., Palmer, J., Ramsey, C. B., Adolphi, F., Muscheler, R., Hughen, K. A., ... & Hogg, A. High-precision dating and correlation of ice, marine and terrestrial sequences spanning Heinrich Event 3: Testing mechanisms of interhemispheric change using New Zealand ancient kauri (Agathis australis). Quaternary Science Reviews, 137, 126-134.https://doi.org/10.1016/j.quascirev.2016.02.005, 2016

Svensson, A., Dahl-Jensen, D., Steffensen, J. P., Blunier, T., Rasmussen, S. O., Vinther, B. M., ... & Bigler, M. Bipolar volcanic synchronization of abrupt climate change in Greenland and Antarctic ice cores during the last glacial period. Climate of the Past, 16(4), 1565-1580. https://doi.org/10.5194/cp-16-1565-2020https://doi.org/10.5194/cp-16-1565-2020, 2020

WAIS Divide Project Members. Precise interpolar phasing of abrupt climate change during the last ice age. Nature, 520(7549), 661-665. https://doi.org/10.1038/nature14401 ,  2015

---

## Author Comment (AC2)

**Reply to Pete D. Akers (Referee) regarding cp-2022-62**

"Synchronizing ice-core and U/Th time scales in the Last Glacial
Maximum using Hulu Cave 14C and new 10Be measurements from Greenland and
Antarctica" by Giulia Sinnl et al., Clim. Past Discuss.,
https://doi.org/10.5194/cp-2022-62-RC2, 2023

We thank the referee Pete D. Akers for the feedback. We have marked our responses to each comment in blue.

>>General comments
>>In this paper, the authors examine the chronologies in three sets of paleo records over the Last Glacial Maximum: Greenland ice cores, a West Antarctic ice core, and Hulu Cave speleothems. To examine the synchroneity of the existing chronologies for each record set, the authors use cosmogenic nuclides as the comparative proxy under the assumption that changes in these nuclide fluxes are globally synchronous and avoid the impacts of local climate that can cause issues when matching other environmental proxies like stable isotopes of oxygen and carbon. They use these comparison to estimate that over a century offset exists between the Greenland and Antarctic chronologies, and both ice core records are offset from the speleothem record by 200 additional years. I found this paper to be very well written and organized. Their approach of linking 10Be with 14C data is well-thought out and impressive in its level of detail dedicated to modelling and cross-examining between records. The paper overall is fairly heavy on methods and technical speak, but such a focus is likely necessary due to the nature of this study, and I applaud the authors for providing very detailed information on all parts of this work. In particular, there is very good supportive data in the captions (e.g., supporting citations, clear descriptions of figure color/symbols), which I often find lacking in papers under review. That said, some captions are very lengthy, and while I think the supportive information is good, some considerations could be made for making the information in the captions more succinct. Similarly, I'd suggest examining if any of the deep technical details could be streamlined without sacrificing important context, which I think would help the paper prove more approachable to a broader audience. However, I think this manuscript still manages to have a clear narrative structure and is one of the better manuscripts that I've reviewed in terms of presenting clear results and conclusions while still providing deep technical detail. My specific comments are minor. Some of the modelling aspects of the paper are outside my scientific expertise, but the described approaches and discussion of the results are all logical to me and supported by ample evidence. Some additional focus could possibly be made toward considering in more detail the impact these new chronologies should have on paleoclimate reconstructions, or how the ice core community should respond to the findings of century-scale offsets in chronologies. Such additional information are not required edits from me, but offered here as consideration to the authors if they wish to elaborate more.
There are many acronyms/abbreviations in the paper. To improve readability, I suggest eliminating the ones that are only used once or twice, which I list in the technical corrections. Throughout the paper, uncertainties in age corrections and estimates are usually given as 1 SD, but a 95% confidence interval would likely be more informative and representative. As of now, the offsets between the different chronology sets are sometimes within the 95% confidence window. I think the magnitude of difference and consistency of difference is large enough to believe you are observing true offsets, but you might want to target this point more with a

dedicated response to avoid the appearance that you used 1 SD values to make the results look more differentiated from each other.

Some restructuring of the introduction would probably make the paper more effective. It takes quite a while before you point out what your paper is doing, and what problem currently exists that you are attempting to solve. So we read about LGM events like stadials and HEs, but without the context for why we need to know this. Likewise, there is a lot of discussion about the specifics of the three different chronologies without the reader's context for why this information will be important later. Being more explicit about the currently known or expected issues in record synchroneity early on and repeatedly linking the background information to these issues and your study approaches will make for a tighter, more effective introduction.

Reply: Thank you for your positive feedback of the overall contents and structure of the manuscript. We will restructure the instruction to be more informative and improve its readability. We will also reflect on the broader impact of our research for the paleoclimate community, as you suggest. We will shorten the captions where necessary and avoid redundant acronyms. As for the uncertainty issue, we have dedicated some discussion about how to possibly reduce them, and we will convert to 95% intervals throughout.

>>Specific comments

>>65: I'm slightly unclear in what this sentence is meaning by "correspondence". The first part makes it clear that HS-1 is only part of GS 2.1, but "a correspondence with the late GS-3" could be taken either as 1) HS-2 is similar to HS-1 in that it only covers part of GS-3 (the late part) or 2) unlike HS-1, HS-2 coincides with the entirety of the late GS-3 unit. A rewording will help the point you are intending to make be clear.

Reply: We meant it as in 1), i.e. we observe a similarity between the two HSs. We will clarify this.

>>83: The objective is determining if the accepted chronologies at three sites actually align when compared with a globally synchronous marker, right? "Comparison of three time scales in the LGM" is vague and doesn't really capture the point of your work.

Reply: Thank you for your point, we have clarified the statement about our objectives.

>> 589: This is probably my biggest comment here, in that all the age offset "fault" is taken to lie with the ice cores. I think that this is reasonable, since the layer counting of ice cores has known issues, but it would still be good to elaborate a little more on what the UTh uncertainties might represent. How much older would the "too old" estimates be, as cited by Corrick? Do any estimates exist for this time period? This might get more into the mechanics of U-Th dating, thorium corrections, U-Th half life estimates, etc, than warranted by your study, but any additional clarification and constraint you can give on this point will make your argument stronger toward the ice core re-dating conclusion.

Reply: Thank you for this comment. The layer identification issues pose a clear challenge to the accuracy of ice core time scales. We are not aware of estimates in the LGM period showing Hulu being dated too old, but Corrick et al. (2020) do mention a possible issue of sub-optimal

sample positioning of U/Th and/or $\delta^{18}$O. If one uses the argument of $\delta^{18}$O synchroneity across Asian speleothems, considering all caveats because of climate effects, then one can observe a spread of the HE-2 onset between Hulu and the Cherrapunji speleothem (Dong et al., 2022) of about 100 years, where the Hulu record shows the oldest onset of HE2. This could indicate that dating issues in Hulu are in fact present (given that Cherrapunji was very carefully counted over the LGM using annual lamina). If Hulu was 100 years too old in the LGM, then the offset to GICC05 and WDC would obviously be smaller. However, there is no radionuclide data for the Cherrapunji speleothem, so our methodology remains strongly dependent on the Hulu $^{14}$C record. We have presented these arguments in the revised manuscript.

>>626: MCE was defined back at line 98 but hasn't been used until now. I had forgotten what this acronym stood for and had to look it up again. It would probably be good to redefine it again here (or around 700 when it is used a lot again) for the reader.

Reply: We will redefine MCE here.

>>Technical corrections

Reply: All technical corrections have been considered and accepted. Thank you for your detailed review.

>>60: Probably worth spelling out Heinrich events here, since it starts the sentences and HEs is only used in this way once here.
>>70: "it has been suggested that during HS-1, the empirical" – removing a comma will make the sentence read better
>>76: Check this sentence for phrasing. I think the first comma needs to go after GS-3, but the sentence does not make full sense at the moment. It is also rather long and run-on.
>>80: Similarly to HEs, I would spell out Heinrich stadials here to aid the reader already seeing lots of abbreviations, and also HSs is only used once here.
>>88: Just a consideration that you could term this 14Ccalcite to be consistent with the other stalagmite proxies.
>>168: The only "previously reported" radionuclide tie point? Or is this the only excursion that can function as a tie point according to some parameter?
>>179: Table should be capitalized. Also Fig. and Table throughout manuscript.
>>187: Period after "Table 1". Also in all other captions for figures and tables as well. May be caught in proofing.
>>Table 1: Geographic coordinates need degrees symbols. All sites should have same level of significant decimal digits (see WDC longitude).
>>192: Timescale here is one word whereas earlier it is written as two words (time scale). Consistency needed (or use "chronology" term instead).
>>Figure 1: Be aware that overlapping symbols in red and green (Fig 1e) may cause accessibility issues for colorblind readers. Check with a colorblind filter to make sure figure color schemes are accessible.
>>Figure 2: The image is fuzzy compared to Figure 1, so just make sure there is a high quality version for final submission.
>>241: "here, the box-diffusion…"

>>264: GCRs not GCR's. Also, consider just spelling this out since it is the only use of this abbreviation in the manuscript.

>>246: The comma after 14C seems oddly placed for phrasing in this sentence.

>>Figure 6: No y axis marks for d, but some superfluous yellow ones at right. Check y-axes labels and numbers for overlap (e.g., a+b, c+d).

>>Figure 8: I appreciate the dedication to information in these captions. This caption, however, is very wordy and could use some of the methods text in it to be greatly summarized or moved into the main text or supplement.

Kind regards,
Giulia Sinnl et al.

References:

Corrick, E. C., Drysdale, R. N., Hellstrom, J. C., Capron, E., Rasmussen, S. O., Zhang, X., ... & Wolff, E. (2020). Synchronous timing of abrupt climate changes during the last glacial period. Science, 369(6506), 963-969.

Dong, X., Kathayat, G., Rasmussen, S. O., Svensson, A., Severinghaus, J. P., Li, H., ... & Cheng, H. (2022). Coupled atmosphere-ice-ocean dynamics during Heinrich Stadial 2. *Nature communications*, *13*(1), 1-14.

---

## Author Response (AR1)

List of all relevant changes made in the manuscript "Synchronizing ice-core and U/Th time scales in the Last Glacial Maximum using Hulu Cave 14C and new 10Be measurements from Greenland and Antarctica"

- Following the request by the reviewers, the introduction was restructured and shortened. Sections too technical were moved to the methods or discussion parts, while sections that were outside the scope of the manuscript were removed.
- The goal of the paper was rephrased in the introduction.
- A new figure 1 was added, in lieu of the table proposed by the 1st referee, to describe the climatic stages of the introduction.
- A reference to ages BP was added to explain the age conversion.
- All figure captions were shortened.
- In the methods, paragraph exceeding the scope of the paper were removed.
- All figures were edited for colour blindness and clarity.
- 95% uncertainty intervals were explicitly mentioned in the results, but we decided to keep the 68% intervals throughout to not undermine the result of the paper.
- We reduced the use of abbreviations.
- We added a comment on the Hulu timescale accuracy in discussion 4.3.
- We remarked on the outlook for the uncertainties in the conclusion paragraph.
- All other minor comments were addressed.

Sinnl et al.